# Pessimistic Model-based Offline Reinforcement Learning under Partial Coverage

**Masatoshi Uehara, Wen Sun**
Department of Computer Science
Cornell University, Ithaca, NY 14850, USA
`{mu223,ws455}@cornell.edu`

## Abstract

We study model-based offline Reinforcement Learning with general function approximation without a full coverage assumption on the offline data distribution. We present an algorithm named *Constrained Pessimistic Policy Optimization (CPPO)* which leverages a general function class and uses a constraint over the model class to encode pessimism. Under the assumption that the ground truth model belongs to our function class (i.e., realizability in the function class), CPPO has a PAC guarantee with offline data only providing partial coverage, i.e., it can learn a policy that competes against any policy that is covered by the offline data. We then demonstrate the flexibility of CPPO: it can be seamlessly applied to many specialized Markov Decision Processes where additional structural assumptions further refine the concept of partial coverage. Two notable examples are: (1) *low-rank MDP with representation learning* where the partial coverage condition is defined using a relative condition number measured by the unknown ground truth feature representation; (2) *factored MDP* where the partial coverage condition is defined using density ratios associated with individual factors.

## 1 Introduction

Offline Reinforcement Learning (RL) is one of the important areas of RL where the learner is presented with a static dataset consisting of transition-related information (state, action, reward, and next state) collected by some behavior policy, and needs to learn purely from the offline data without any future online interaction with the environment. Offline RL is used in a number of applications where online random experimentation is costly or dangerous such as health care (Kosorok & Laber, 2019), digital marketing (Chen et al., 2019) and robotics (Levine et al., 2020).

The performance guarantees of offline RL often rely on two quantities: the coverage of the offline data and the property of the function approximation used in the algorithms. For instance, for the classic Fitted-Q-iteration (FQI) algorithm (Ernst et al., 2005; Munos & Szepesvári, 2008), it requires (a) full coverage in the offline data, i.e., $\max_{(s,a)} d^\pi(s,a)/\rho(s,a) < \infty$ for *any* stochastic policies $\pi$ including history-dependent non-Markovian policies, where $d^\pi(s,a)$ is a state-action occupancy distribution of a policy $\pi$ and $\rho(s,a)$ is an offline distribution, (b) realizability in a Q function class, i.e., the optimal Q function belongs to the function class, and (c) Bellman completeness, i.e., applying the Bellman operator on any function in the function class results in a new function that also belongs to the function class (see the first row in Table 1). Among these three assumptions, the full coverage and the Bellman completeness are particularly strong. The full coverage means that the behavior policy needs to be exploratory enough, although figuring out an exploratory policy itself is an extremely hard problem for large-scale MDPs. The Bellman completeness assumption does not have a monotonic property, i.e., even starting with a function class that originally permits Bellman completeness, slightly increasing the capacity of the function class could result in a new class that does not have Bellman completeness anymore. Thus, we aim to relax the assumptions on the offline data and the function class. Particularly, we are interested in the following question:

> Given a realizable function class and an offline distribution that only provides partial coverage, can we learn a policy that is able to compete with any policy that is covered by the offline distribution?

*We study this question from a model-based learning perspective and provide an affirmative answer to the question.* More specifically, different from FQI, we start with a realizable model class, i.e., the ground truth transition falls into the model class. We further abandon the strong full coverage assumption, and instead, assume partial coverage which means the offline data distribution only covers a state-action distribution of *some high-quality comparator policy* $\pi^*$ ($\pi^*$ is not necessarily

| Methods | Type | Coverage | Additional Structures |
|---|---|---|---|
| FQI (Munos & Szepesvári, 2008) | **F** | Full: $\max_{s,a} \frac{d^\pi(s,a)}{\rho(s,a)} < \infty, \forall \pi$ | Bellman complete |
| Minimax Way (Uehara et al., 2020) | **F** | Full: $\max_{s,a} \frac{d^\pi(s,a)}{\rho(s,a)} < \infty, \forall \pi$ | Realizability in density ratio |
| Duan et al. (2020) | **F** | Full: $\mathbb{E}_{s,a\sim\rho}\phi(s,a)\phi(s,a)^\top$ is PSD | Linear Bellman complete |
| Xie & Jiang (2020) | **F** | Full: $\max_{s,a,s'} \frac{P^*(s'|s,a)}{\rho(s')} < \infty$ | None |
| Liu et al. (2020) | **F** | Partial$^\dagger$ : $\max_{s,a} \frac{d^{\pi^*}(s,a)}{\rho(s,a)} < \infty$ | Bellman / Policy class complete |
| Rashidinejad et al. (2021) | **F** | Partial: $\max_{s,a} \frac{d^{\pi^*}(s,a)}{\rho(s,a)} < \infty$ | Tabular MDP |
| Jin et al. (2020b); Zhang et al. (2021b) | **F** | Partial$^{\dagger\dagger}$: $\max_x \frac{x^\top \mathbb{E}_{s,a\sim d^{\pi^*}} \phi(s,a)(\phi(s,a))^\top x}{x^\top \mathbb{E}_{s,a\sim\rho}\phi(s,a)(\phi(s,a))^\top x} < \infty$ | Linear MDP (Jin et al., 2020a) |
| Xie et al. (2021) | **F** | Partial: $\max_f \frac{\|f-\mathcal{T}f\|^2_{d^{\pi^*}}}{\|f-\mathcal{T}f\|^2_\mu} < \infty$ | Bellman complete |
| Zanette et al. (2021) | **F** | Partial : $\max_x \frac{x^\top \mathbb{E}_{s,a\sim d^{\pi^*}} \phi(s,a)(\phi(s,a))^\top x}{x^\top \mathbb{E}_{s,a\sim\rho}\phi(s,a)(\phi(s,a))^\top x} < \infty$ | Linear Bellman complete |
| Batch (Ross & Bagnell, 2012) | **B** | Full: $\max_{s,a} \frac{d^\pi(s,a)}{\rho(s,a)} < \infty, \forall \pi$ | None |
| Milo (Chang et al., 2021) | **B** | Partial: $\max_x \frac{x^\top \mathbb{E}_{s,a\sim d^{\pi^*}} \phi(s,a)(\phi(s,a))^\top x}{x^\top \mathbb{E}_{s,a\sim\rho}\phi(s,a)(\phi(s,a))^\top x} < \infty$ | KNR / GP |
| CPPO (Ours) | **B** | Partial$^{\dagger\dagger\dagger}$: $\max_{s,a} \frac{d^{\pi^*}(s,a)}{\rho(s,a)} < \infty$ 
 Partial: $\max_x \frac{x^\top \mathbb{E}_{s,a\sim d^{\pi^*}} \phi(s,a)(\phi(s,a))^\top x}{x^\top \mathbb{E}_{s,a\sim\rho}\phi(s,a)(\phi(s,a))^\top x} < \infty$ 
 Partial: $\max_x \frac{x^\top \mathbb{E}_{s,a\sim d^{\pi^*}} \phi^*(s,a)(\phi^*(s,a))^\top x}{x^\top \mathbb{E}_{s,a\sim\rho}\phi^*(s,a)(\phi^*(s,a))^\top x} < \infty$ | None 
 Linear MDP /KNR / GP 
 Low-rank MDP (unknown $\phi^*$) |

Table 1: Comparison among existing works regarding their type, coverage, and additional structural assumptions on the function class or MDPs. Type **F** means model-free and type **B** means model-based. Partial coverage means [2] that the offline distribution $\rho$ covers a state-action distribution of a comparator policy $\pi^*$. $\dagger$ means it assumes an accurate density estimator for $\rho(s,a)$. $\dagger\dagger$ means although the analysis in Jin et al. (2020a) is done under the full coverage for linear MDPs, based on the argument (Zhang et al., 2021b), we can show the algorithm has the PAC guarantee under partial coverage in terms of the relative condition number for linear MDPs. $\dagger\dagger\dagger$ means that we can refine it to a more adaptive quantity using the model class (i.e., Definition 1). All the methods in the table require realizability in the function class.

the optimal policy, and $\pi^*$ could be non-Markovian), i.e., $\max_{s,a} d^{\pi^*}(s,a)/\rho(s,a) < \infty$, We design an algorithm — Constrained Pessimistic Policy Optimization (CPPO), which can learn a policy that is as good as any comparator policy $\pi^*$ that is covered by the offline data. The fact that CPPO can learn to compete against history-dependent policies is meaningful in offline RL when the offline data does not cover the optimal policy.

While one could assume density ratio based concentrability coefficient ($\max_{s,a} d^{\pi^*}(s,a)/\rho(s,a)$) to be under control for small size MDPs, in large-scale MDPs (e.g. continuous state space), the density ratio could quickly become an extremely large quantity which makes the performance guarantee vacuous. When applying CPPO to MDPs with additional structural assumptions, we can seamlessly refine the density ratio based concentrability coefficient to more natural and tighter quantities. Notably, we consider the offline representation learning setting where the underlying MDPs permit a low-rank structure (unlikely linear MDPs (Jin et al., 2020a; Yang & Wang, 2020), we do not assume the ground truth state-action feature representation $\phi^\star$ is known, and instead we need to learn $\phi^\star$) and we show that we can refine the density ratio to a relative condition number that is defined using the *unknown* true state-action feature representation $\phi^\star$. Intuitively this means that as long as there exists a high-quality comparator policy that only visits the subspace (defined using the true representation $\phi$) that is covered by the offline data, CPPO can compete against such a policy, *even without knowing the true* $\phi^\star$. Such bounded relative condition number assumption is much weaker than the bounded density ratio assumption.[3] While the concept of relative condition number was originally introduced in the online RL setting (e.g., Agarwal et al. (2020c;a) with a known linear feature $\phi$), and later was introduced in offline RL (Zhang et al. (2021b); Chang et al. (2021)), these prior works *all* rely on the fact that the feature representation $\phi$ is known to the learner a priori (see Table 1 for the comparison). Another interesting example is factored MDPs (Kearns & Koller, 1999) where we show CPPO refines the density ratios to be density ratio associated with individual factors, which leverages the factored structure and is provably tighter. We also give examples on linear MDPs (Yang & Wang, 2020), kernelized nonlinear regulator (KNRs) (Kakade et al., 2020), where we again show that CPPO enjoys problem specific quantities for measuring the coverage.

**Our contributions.** Our contributions are two folds, which we summarize below:

---

[3]Strictly speaking, in Jin et al. (2020b); Rashidinejad et al. (2021), a comparator policy is restricted to the optimal policy. In Chang et al. (2021); Zanette et al. (2021); Xie et al. (2021) and CPPO, a comparator policy can be any policy.

1. We show that in the model-based setting, realizability and partial coverage is enough to learn a high-quality comparator policy (Theorem 1). Notably, (1) this result holds for *any MDPs with realizable model classes*, (2) we can compete against even history-dependent policies. This is in sharp contrast to the state-of-art provable model-free offline RL results: see Table 1 on page 2 for detailed comparisons to prior works.

2. Under additional structural assumptions (e.g., KNRs, linear MDPs (Yang & Wang, 2020), linear mixture MDPs (Ayoub et al., 2020), low-rank MDPs, factored MDPs), we show that we can seamlessly refine the density ratio based concentrability coefficients to problem specific quantities. This flexibility to adapt to problem specific coverage measuring quantities is in sharp contrast to other model-free offline RL algorithms such as minimax based approaches (Uehara et al., 2020) which, to the best of our knowledge, cannot leverage MDP's structures (e.g., linear MDPs) to refine its density ratio based concentrability coefficients.

While we focus on the model-based setting and have demonstrated advantages of our approach over model-free ones (i.e., no more Bellman completeness assumption on function classes, being able to compete against a larger pool of policies, and the ability to seamlessly adapt to problem dependent structures), it is worth noting that realizability in the model-based setting is usually considered stronger than the one in the model-free setting. On the empirical side, model-based offline RL algorithms are the state-of-art (e.g., Yu et al. (2020); Kidambi et al. (2020); Matsushima et al. (2020); Cang et al. (2021); Chang et al. (2021)). Our theoretical results provide a sharp contrast between model-based and model-free approaches in offline RL.

## 2 RELATED WORK

We discuss two families of related works of offline RL. In Appendix C, we discuss related works about representation learning in RL.

Insufficient coverage of the dataset due to the lack of online exploration is known as the main challenge in offline RL (Wang et al., 2020). To deal with this problem, a number of methods have been recently proposed from both model-free (Wu et al., 2019; Touati et al., 2020; Kumar et al., 2020; Liu et al., 2020; Rezaeifar et al., 2021; Fujimoto et al., 2019; Fakoor et al., 2021; Ghasemipour et al., 2021; Buckman et al., 2020) and model-based perspectives (Yu et al., 2020; Kidambi et al., 2020; Matsushima et al., 2020; Yin et al., 2021). More or less, their methods rely on the idea of pessimism and its variants in the sense that the learned policy can avoid uncertain regions not covered by offline data. As a theoretical side, Munos & Szepesvári (2008); Duan et al. (2020; 2021); Fan et al. (2020) proved FQI has a PAC (probably approximately correct) guarantee under realizability, the global coverage, and Bellman completeness conditions. Other offline model-free RL methods such as minimax offline RL methods also require realizability and the global coverage (Chen & Jiang, 2019; Antos et al., 2008; Uehara et al., 2021; Duan et al., 2021; Zhang et al., 2020; Nachum et al., 2019). Recently, by leveraging the aforementioned the pessimism idea, Jin et al. (2020a); Rajaraman et al. (2020) showed that pessimistic FQI can be applied to partial coverage setting for linear and tabular MDPs. Comparing to their works, our analysis focuses on model-based approaches with general function approximation. The offline model-based method is known to have a PAC guarantee under the realizability and the global coverage (Ross & Bagnell, 2012; Chen & Jiang, 2019). As the most closely related work, Chang et al. (2021) proved a model-based method with an additional penalty term can weaken the assumption from the global coverage to the partial coverage for structured MDPs such as KNRs and Gaussian Processes models (Deisenroth & Rasmussen, 2011). In this work, we consider *arbitrary* MDPs with a realizable model class and aim for PAC bounds under a partial coverage condition.

## 3 PRELIMINARIES

We consider a Markov Decision process (MDP) $\mathcal{M} = \{\mathcal{S}, \mathcal{A}, P, \gamma, r, d_0\}$ where $P : \mathcal{S} \times \mathcal{A} \to \Delta(\mathcal{S})$ is the transition, $r : \mathcal{S} \times \mathcal{A} \to [0, 1]$ is the reward function, $\gamma \in [0, 1)$ is the discount factor, and $d_0 \in \Delta(\mathcal{S})$ is the initial state distribution. A policy $\pi$ maps from state (or history) to distribution over actions. Given a policy $\pi$ and a transition distribution $P$, $V_P^\pi$ denotes the expected cumulative reward of $\pi$ under $P, d_0$ and $r$. Similarly, $Q_P^\pi : \mathcal{S} \times \mathcal{A} \to \mathbb{R}, A_P^\pi : \mathcal{S} \times \mathcal{A} \to \mathbb{R}$ are a Q-function and advantage-function under $P$ and $\pi$. Given a transition $P$, we denote $\pi(P)$ as the optimal policy associated with model $P$ under reward $r$. We also denote $d_P^\pi \subset \Delta(\mathcal{S} \times \mathcal{A})$ as the average state-action distribution of $\pi$ under the transition model $P$, i.e, $d_P^\pi = (1 - \gamma) \sum_{t=0}^\infty \gamma^t d_{P,t}^\pi$,

where $d_{P,t}^{\pi} \in \Delta(\mathcal{S} \times \mathcal{A})$ is the distribution of $(s_t, a_t)$ under $\pi$ and $P$ at a time-step $t$. We denote the true transition distribution as $P^{\star}$, which we do not know in advance. For simplicity, we suppose $r$ is known. The extension to the unknown reward is straightforward.

In the offline RL setting, we have an offline distribution $\rho \in \Delta(\mathcal{S} \times \mathcal{A})$, and an offline dataset $\mathcal{D} = \{s^{(i)}, a^{(i)}, r^{(i)}, s'^{(i)}\}_{i=1}^n$ which is sampled in the following way: $s, a \sim \rho, r = r(s,a), s' \sim P^{\star}(\cdot|s,a)$. We hope to obtain $\pi(P^{\star}) = \arg\max_{\pi} V_{P^{\star}}^{\pi}$ from this offline dataset without any further interaction with the environment. We often denote $\mathbb{E}_{\mathcal{D}}[f(s,a,s')] = 1/n \sum_{(s,a,s') \in \mathcal{D}} f(s,a,s')$. Our goal is to construct an offline RL algorithm Alg, which maps from $\mathcal{D}$ to $\pi$ so that the suboptimality gap $V_{P^{\star}}^{\pi^*} - V_{P^{\star}}^{\text{Alg}(\mathcal{D})}$ for any comparator policy $\pi^* \in \Pi$ is minimized, where $\Pi$ in this work can be an unrestricted policy class (e.g., including non-Markovian policies). Hereafter, $c, c_1, c_2, \cdots$ are always universal constants.

**Partial coverage.** Throughout this work, we do not assume $\rho$ has global coverage. The global coverage in this work means that the density ratio based concentrability coefficient $d_{P^{\star}}^{\pi}(s,a)/\rho(s,a)$ is upper-bounded by some constant $C \in \mathbb{R}^+$ for all polices $\pi \in \Pi$, or the feature covariance matrix corresponding to the offline distribution $\mathbb{E}_{s,a\sim\rho}\phi(s,a)\phi(s,a)^{\top}$ ($\phi \in \mathcal{S} \times \mathcal{A} \to \mathbb{R}$ is a feature representation) is full rank and has a non-zero minimum eigenvalue, which are commonly used assumptions in offline RL (Munos, 2005; Antos et al., 2008; Chen & Jiang, 2019; Duan et al., 2020). Under the full coverage, they show the output policy can compete with the globally optimal policy $\pi(P^{\star})$. However, this assumption may not be true in practice as computing an exploratory policy itself is a challenging task for large-scale RL problems. Instead, we are interested in the partial coverage setting such as $d_{P^{\star}}^{\pi^*}(s,a)/\rho(s,a) \leq C$, which means the state-action occupancy measure under some comparator policy $\pi^*$ is covered by the offline dataset. We want to design an algorithm that can compete against *any* policy $\pi^*$ that is covered by the offline data. This assumption is much weaker than the global coverage.

## 4 PESSIMISTIC MODEL-BASED OFFLINE RL

We first introduce a general model-based algorithm that has a PAC guarantee of the suboptimality gap under partial coverage defined with a newly introduced concentrability coefficient. The algorithm takes a realizable model class as input and outputs a policy that is as good as any comparator policy that is covered by the offline data in the sense of the bounded concentrability coefficient.

Our algorithm, *Constrained Pessimistic Policy Optimization (CPPO)* (Algorithm 1), takes a realizable hypothesis class $\mathcal{M}$ (with $P^{\star} \in \mathcal{M}$) consisting of $|\mathcal{M}|$ candidate models as input, computes the maximum likelihood estimator (MLE) $\widehat{P}_{\text{MLE}}$ using the given offline data $\mathcal{D} = \{s, a, s'\}$. It then forms a min-max objective subject to a constraint. The min-max objective introduces pessimism via searching for the least favorable model $P$ (in terms of its policy's value $V_P^{\pi}$) that is feasible with respect to the constraint. We can also express the constrained optimization procedure using a version space $\mathcal{M}_{\mathcal{D}}$ and a policy optimization procedure defined below:

$$\max_{\pi \in \Pi} \min_{P \in \mathcal{M}_{\mathcal{D}}} V_P^{\pi}, \quad \text{where } \mathcal{M}_{\mathcal{D}} = \left\{ P \mid P \in \mathcal{M}, \mathbb{E}_{\mathcal{D}}\left[\text{TV}(\widehat{P}_{\text{MLE}}(\cdot|s,a), P(\cdot|s,a))^2\right] \leq \xi \right\}, \quad (1)$$

where $\text{TV}(P_1, P_2)$ is a total variation (TV) distance between two distributions $P_1$ and $P_2$. The version space $\mathcal{M}_{\mathcal{D}}$ contains models that are not far away from $\widehat{P}_{\text{MLE}}$ in terms of the average TV distance under $\mathcal{D}$. The version space is constructed such that with high probability $P^{\star} \in \mathcal{M}_{\mathcal{D}}$.

Below we state the algorithm's performance guarantee. Assuming for now that $P^{\star} \in \mathcal{M}_{\mathcal{D}}$ holds with high probability, then, $\hat{V}^{\pi} := \min_{P \in \mathcal{M}_{\mathcal{D}}} V_P^{\pi}$ is a pessimistic policy evaluation estimator, which satisfies $\hat{V}^{\pi} \leq V_{P^{\star}}^{\pi}$ for all $\pi \in \Pi$. Using the idea of pessimism, we have the following observation:

$$V_{P^{\star}}^{\pi^*} - V_{P^{\star}}^{\hat{\pi}} = V_{P^{\star}}^{\pi^*} - \hat{V}^{\pi^*} + \hat{V}^{\pi^*} - V_{P^{\star}}^{\hat{\pi}} \leq V_{P^{\star}}^{\pi^*} - \hat{V}^{\pi^*} + \hat{V}^{\hat{\pi}} - V_{P^{\star}}^{\hat{\pi}} \leq V_{P^{\star}}^{\pi^*} - \hat{V}^{\pi^*},$$

where the first inequality uses $\hat{\pi} = \arg\max_{\pi \in \Pi} \hat{V}^{\pi}$ and the second inequality uses $\hat{V}^{\pi} \leq V_{P^{\star}}^{\pi}$ for all $\pi \in \Pi$. Thus, the final error only incurs the policy evaluation error for the comparator policy $\pi^*$, which leads to the error only depending on the concentrability coefficient for the comparator policy.

We define the following new concentrability coefficient that uses the model class $\mathcal{M}$ :

---

**Algorithm 1** Constrained Pessimistic Policy Optimization (CPPO)

1: **Require**: Models $\mathcal{M}$, dataset $\mathcal{D}$, parameter $\xi$, policy class $\Pi$ (note $\Pi$ could be unrestricted)
2: Obtain the estimator $\hat{P}_{\text{MLE}}$ by MLE: $\hat{P}_{\text{MLE}} = \arg\max_{P \in \mathcal{M}} \mathbb{E}_{\mathcal{D}}[\ln P(s' \mid s, a)]$.
3: Constrained policy optimization:

$$\hat{\pi} = \arg\max_{\pi \in \Pi} \min_{P \in \mathcal{M}} V_P^\pi, \text{ s.t., } \mathbb{E}_{\mathcal{D}}\left[\text{TV}(\hat{P}_{\text{MLE}}(\cdot|s,a), P(\cdot|s,a))^2\right] \le \xi.$$

4: **Return** $\hat{\pi}$

---

**Definition 1** (Model-based Concentrability Coefficient). *For a comparator policy $\pi^*$, we define the concentrability coefficient $C_{\pi^*}^\dagger$ as follows:*

$$C_{\pi^*}^\dagger = \sup_{P' \in \mathcal{M}} \frac{\mathbb{E}_{(s,a) \sim d_{P^\star}^{\pi^*}}[\text{TV}(P'(\cdot|s,a), P^\star(\cdot|s,a))^2]}{\mathbb{E}_{(s,a) \sim \rho}[\text{TV}(P'(\cdot|s,a), P^\star(\cdot|s,a))^2]}.$$

The following theorem shows CPPO learns a policy that competes against $\pi^*$ when $C_{\pi^*}^\dagger < \infty$.

**Theorem 1** (PAC Bound for CPPO with general function class). *Assume $P^\star \in \mathcal{M}$. We set $\xi = c_1 \frac{\ln(c_2 |\mathcal{M}|/\delta)}{n}$. Then, with probability $1 - \delta$, for any comparator policy $\pi^* \in \Pi$ ($\Pi$ can be the unrestricted policy class containing non-Markovian policies),*

$$V_{P^\star}^{\pi^*} - V_{P^\star}^{\hat{\pi}} \le c_3(1-\gamma)^{-2}\sqrt{\frac{C_{\pi^*}^\dagger \ln(c_2|\mathcal{M}|/\delta)}{n}}.$$

To the best of our knowledge, this is the *first* algorithm that achieves a PAC guarantee for *any MDPs* under the partial coverage assumption $C_{\pi^*}^\dagger < \infty$ with only a realizable hypothesis class. We emphasize that the inequality in the above *uniformly* holds for *all* policies with probability $1 - \delta$ including *history-dependent non-Markovian policies* (see Remark 2). Note that the ability to compete against non-Markovian policies in offline RL is meaningful when the offline data does not cover the optimal policy $\pi^\star$ (i.e., there could be a high-quality history-dependent policy that is covered by the offline data against which we want to compete). In model-free approaches, this type of result generally cannot be obtained. Indeed, the model-free approach from Xie et al. (2021) requires $\Pi$ to be a restricted Markovian policy class, since their bound contains $\text{poly}(\ln(|\Pi|))$ dependence. For the detailed discussion, refer to Remark 1.

The quantity $C_{\pi^*}^\dagger$ adaptively captures the discrepancy between the offline data and the state-action occupancy measure under a comparator policy $\pi^*$ depending on the model class $\mathcal{M}$. For example, $C_{\pi^*}^\dagger$ can be reduced to a relative condition number in KNRs. Besides, it is always upper bounded by the density ratio based concentrability coefficient:

$$C_{\pi^*,\infty} := \sup_{(s,a)} \frac{d_{P^\star}^{\pi^*}(s,a)}{\rho(s,a)}.$$

One extreme case is that functions in $\mathcal{M}$ are all the same, which implies $C_{\pi^*}^\dagger = 1$ regardless.

Theorem 1 consider the case where the hypothesis class $\mathcal{M}$ is finite. When the hypothesis class is infinite, we can still obtain the PAC guarantee by utilizing the generalized result in Section A for any realizable model class with valid statistical complexity (e.g., localized Rademacher complexity).

Prior works that achieve PAC guarantees with only realizable model classes rely on much stronger global coverage $\sup_\pi C_{\pi,\infty} < \infty$ (Chen & Jiang, 2019). Even when the comparator policy is the optimal policy $\pi(P^\star)$, the partial coverage condition $C_{\pi(P^\star),\infty} < \infty$ is weaker. Existing pessimistic model-based algorithms and their theoretical results (Chang et al., 2021) often assume that a *point-wise* model uncertainty measure is given as a by-product of model fitting, which limits the applicability to special linear models such as KNRs/GPs. CPPO can work for any MDPs with the realizable function class having a valid statistical complexity such that the MLE properly works.

**Remark 1** (Comparison to the model-free approach from Xie et al. (2021); Zanette et al. (2021)). *Xie et al. (2021) study the model-free setting where the function class $\mathcal{Q}$ models $Q$ functions assumed to be Bellman complete for any Markovian policy in $\Pi$. While directly comparing model-based approaches to model-free approaches is hard as they use different inductive biases in function classes, we can leverage the approach from Chen & Jiang (2019, Corollary 6) to convert a model*

*class $\mathcal{M}$ to a pair of $\mathcal{Q}$ and $\Pi$ class. Specifically, we can convert a model class $\mathcal{M}$ to a pair of $\mathcal{Q}$ class and $\Pi$ class such that $\mathcal{Q}$ will be realizable and also Bellman complete with respect to all $\pi \in \Pi$. After such conversion from the model-based setting to the model-free setting, running the algorithm from Xie et al. (2021) using $\mathcal{Q}$ and $\Pi$ achieves $V_{P^\star}^{\pi^*} - V_{P^\star}^{\hat{\pi}} = \sqrt{C^\diamond \ln(|\mathcal{M}||\Pi|/n)}, \forall \pi^* \in \Pi$, where $C^\diamond$ is some concentrability coefficient. For the detailed derivation, we refer readers to Appendix D. Since the suboptimality gap from such conversion incurs $\log |\Pi|$, a policy class $\Pi$ cannot be too large. Especially, unlike our results, it cannot take the unrestricted policy class as $\Pi$. This restriction cannot be fixed even if we use natural policy gradient (NPG) algorithms unless models have special structures (Xie et al., 2021; Zanette et al., 2021). The details are given in Section D.*

Thus, our theorem indicates two advantages of model-based approaches: (1) realizability in function class is enough to ensure a PAC guarantee under a partial coverage condition, (2) it can compete against a larger pool of candidate policies including history-dependent non-Markovian policies, which is a meaningful property when the offline data does not cover the globally optimal policy. Next, we demonstrate another key advantage of our approach which is its flexibility to be seamlessly applied to MDPs with special structures.

## 5 EXAMPLES WITH REFINED CONCENTRABILITY COEFFICIENTS

In the previous section, our results apply to any MDP as long as its true transition belongs to a function class $\mathcal{M}$. In this section, we consider several concrete MDPs with additional structural conditions. We show that by leveraging the additional structural conditions, we can refine the model-based concentrability coefficient to more natural quantities. The examples that we discuss here are: (1) linear mixture MDPs which generalize linear MDPs from Yang & Wang (2020) and tabular MDPs, (2) KNRs which generalize LQRs, (3) low-rank MDPs, and (4) factored MDPs.

Before proceeding, we clarify CPPO cannot capture linear MDPs in Jin et al. (2020a) that is different from the one (Yang & Wang, 2020) we use, and linear Bellman-complete MDPs (Duan et al., 2020) without any modification since MLE-based model learning is no longer applicable to them. However, other objective functions for learning models could be applied to these models (e.g., see the nonparametric model-based learning approach from Lykouris et al. (2021); Neu & Pike-Burke (2020) in the online setting), which we leave it as a future work.

### 5.1 TABULAR MDPS AND LINEAR MIXTURE MDPS

**Tabular MDPs** Tabular MDPs are MDPs where the state and action spaces are finite. Although the corresponding hypothesis class for tabular MDPs is infinite, we can still run MLE, that is, estimating $P^\star$ by the empirical distribution. Then, Algorithm 1 has the following guarantee.

**Corollary 1** (PAC bound for tabular MDP). *We set $\xi = c_1 \frac{|\mathcal{S}|^2 |\mathcal{A}| \ln(n|\mathcal{S}||\mathcal{A}|c_2/\delta)}{n}$. Then with probability $1 - \delta$, for all $\pi^* \in \Pi$,*

$$V_{P^\star}^{\pi^*} - V_{P^\star}^{\hat{\pi}} \leq c_3(1-\gamma)^{-2} \left\{ \sqrt{\frac{C_{\pi^*,\infty} |\mathcal{S}|^2 |\mathcal{A}| \ln(n|\mathcal{S}||\mathcal{A}|c_4/\delta)}{n}} \right\}.$$

Here, for tabular MDPs with $\mathcal{M} = \{P : P(\cdot|s,a) \in \Delta(\mathcal{S}), \forall s, a\}$, the model-based concentrability coefficient in Definition 1 is equal to the density ratio based concentrability coefficient $C_{\pi^*,\infty}$ which is the right quantity for small-size tabular MDPs.

**Linear mixture MDPs** We define linear mixture MDPs (Ayoub et al., 2020; Modi et al., 2020).

**Definition 2** (Linear mixture MDPs). *Given a feature vector $\psi : (\mathcal{S}, \mathcal{A}, \mathcal{S}) \to \mathbb{R}^d$, a linear mixture MDP is an MDP where the ground truth transition is $P^\star(s'|s,a) := \theta^{\star\top}\psi(s,a,s'), \theta^\star \in \mathbb{R}^d$.*

By setting, $\psi(s,a,s') = \mu(s') \bigotimes \phi(s,a)$ ($\otimes$ denotes the Kronecker product), linear mixture MDPs include the following linear MDPs (Yang & Wang, 2020):

**Definition 3** (Linear MDPs). *Linear MDP has $P^\star(s'|s,a) := \sum_{i=1}^{d_1} \sum_{j=1}^{d_2} M_{ij}^\star \mu_i(s')\phi_j(s,a)$ with $\mu : \mathcal{S} \to \mathbb{R}^{d_1}$ and $\phi : \mathcal{S} \times \mathcal{A} \to \mathbb{R}^{d_2}$ are known features, and $M^\star \subset \mathbb{R}^{d_1 \times d_2}$.*

We use CPPO to learn on linear mixture MDPs. The corresponding $\mathcal{M}$ is

$$\mathcal{M}_{\text{Mix}} = \left\{ \theta^\top \psi(s,a,s') \mid \theta \in \Theta \subset \mathbb{R}^d, \int \theta^\top \psi(s,a,s') \mathrm{d}(s') = 1 \quad \forall(s,a) \right\}.$$

Given a function $V : \mathcal{S} \to \mathbb{R}$, define the state-action feature indexed by $V$ as $\psi_V(s,a) := \int \psi(s,a,s')V(s')\mathrm{d}(s')$, we have the following PAC guarantee.

**Corollary 2** (PAC bound for linear mixture MDPs). *Suppose* $\inf_{s,a,s'} P^\star(s' \mid s,a) \geq c_3 > 0$, $\Theta = \{\theta : \|\theta\|_2 \leq R\}$, $\|\psi_V(s,a)\|_2 \leq 1, \forall V \in \mathcal{S} \to [0,1]$ *and* $P^\star \in \mathcal{M}_{Mix}$. *We set* $\xi = c_1 d \ln^2(c_2 nR/\delta)/n$. *Then, with probability* $1 - \delta$, *for any* $\pi^*$ *in* $\Pi$ *(again* $\Pi$ *can be the unrestricted policy class), CPPO outputs a policy* $\hat{\pi}$ *such that:*

$$V_{P^\star}^{\pi^*} - V_{P^\star}^{\hat{\pi}} \leq c_4(1-\gamma)^{-2}\sqrt{\min(dC_{\pi^*}^\dagger, d^2\bar{C}_{\pi^*,\mathrm{mix}})\frac{\ln^2(c_5 nR/\delta)}{n}}, \qquad (2)$$

*where the concentrability coefficient* $\bar{C}_{\pi^*,\mathrm{mix}}$ *is defined as:*

$$\bar{C}_{\pi^*,\mathrm{mix}} := \sup_{P \in \mathcal{Z}_{P^\star}} \sup_{x \in \mathbb{R}^d} \left( \frac{x^\top \Sigma_{\pi^*,\psi_{V_P^{\pi^*}}} x}{x^\top \Sigma_{\rho,\psi_{V_P^{\pi^*}}} x} \right)$$

*with the localized class* $\mathcal{Z}_{P^\star} := \{P : \mathbb{E}_{(s,a)\sim\rho}[\mathrm{TV}(P(\cdot \mid s,a), P^\star(\cdot \mid s,a))^2] \leq \xi\}$, $\Sigma_{\rho,\psi_{V_P^{\pi^*}}} = \mathbb{E}_{(s,a)\sim\rho}[\psi_{V_P^{\pi^*}}(s,a)\psi_{V_P^{\pi^*}}(s,a)^\top]$, *and* $\Sigma_{\pi^*,\psi_{V_P^{\pi^*}}} = \mathbb{E}_{s,a\sim d_{P^\star}^{\pi^*}}[\psi_{V_P^{\pi^*}}(s,a)\psi_{V_P^{\pi^*}}(s,a)^\top]$.

*When specializing to linear MDPs, the above bound still holds with* $\bar{C}_{\pi^*,\mathrm{mix}}$ *being replaced by the relative condition number* $\bar{C}_{\pi^*}$:

$$\bar{C}_{\pi^*} := \sup_{x\in\mathbb{R}^d} \frac{x^T\Sigma_{\pi^*}x}{x^\top\Sigma_\rho x}, \ where \ \Sigma_\rho = \mathbb{E}_{(s,a)\sim\rho}[\phi(s,a)\phi(s,a)^\top], \ \Sigma_{\pi^*} = \mathbb{E}_{(s,a)\sim d_{P^\star}^{\pi^*}}[\phi(s,a)\phi(s,a)^\top].$$

This is the first PAC-guarantee result in the offline setting under partial coverage $\bar{C}_{\pi^*,\mathrm{mix}} < \infty$ for linear mixture MDPs. $\bar{C}_{\pi^*,\mathrm{mix}}$ is a newly-introduced concentrability coefficient for linear mixture MDPs. This coefficient is measured on the integrated feature vectors $\phi_V(s,a)$ for $V : S \to [0,1]$. Note the class of $V$ is localized, i.e., we consider state-value functions $V_P^{\pi^*}(s)$ for all $P$ centered around $P^\star$ under data distribution $\rho$ (i.e., $P \in \mathcal{Z}_{P^\star}$). Such localization property ensures that $\bar{C}_{\pi^*,\mathrm{mix}} \leq C_{\pi^*}^\dagger$ (see Lemma 10 in Section F).

Note that these relative condition number based quantifiers are always tighter than the density ratio based concentrability coefficients (i.e., $\max\{\bar{C}_{\pi^*}, \bar{C}_{\pi^*,\mathrm{mix}}\} \leq C_{\pi^*,\infty}$). For the special case where $\phi(s,a)$ is a one-hot encoding vector, then they are reduced to the density ratio based concentrability coefficient. In a non-tabular setting, even if when the density ratio is infinite, the relative condition number can be still finite. Intuitively, the bounded relative condition number implies that the offline data covers the subspace that the comparator policy $\pi^*$ visits.

We remark $P^\star(s' \mid s,a) \geq c_3 > 0$ in Corollary 2 is a technical condition that allows us to calculate the entropy integral of the hypothesis class easily. It can be potentially discarded by a more careful argument following (van de Geer, 2000, Chapter 7). The norm assumption $\|\psi_V(s,a)\|_2 < 1$ is commonly assumed in the online setting (Zhou et al., 2021).

## 5.2 Kernelized Nolinear Regulators

We consider the example of KNRs in this section. A kernelized Nonlinear Regulator (KNR) (Kakade et al., 2020) is a model where the ground truth transition $P^\star(s'|s,a)$ is defined as $s' = W^\star\phi(s,a)+\epsilon$, $\epsilon \sim \mathcal{N}(0,\zeta^2\mathbf{I})$, with $\phi : \mathcal{S} \times \mathcal{A} \to \mathbb{R}^d$ being a possibly nonlinear feature mapping. We denote the corresponding model on $W$ by $P(W)$. We can apply Algorithm 1 and obtain its guarantee. Especially, since $\mathrm{TV}(P(W)(\cdot \mid s,a), P(W^\star)(\cdot \mid s,a))^2 = \Theta(\|(W - W^\star)\phi(s,a)\|_2^2)$ (Devroye et al., 2018), $C_{\pi^*}^\dagger$ is upper-bounded by the relative condition number $\bar{C}_{\pi^*}$.

Then, we can also recover the result of Chang et al. (2021) which proposes a reward penalty-based pessimistic offline RL algorithm. The detail is given in Section B. In summary, we can show

$$V_{P^\star}^{\pi^*} - V_{P^\star}^{\hat{\pi}} \leq c_1(1-\gamma)^{-2}\min(d^{1/2}, \bar{R})\sqrt{\bar{R}}\sqrt{\frac{d_\mathcal{S}\bar{C}_{\pi^*}\ln(1+n)}{n}}, .$$

where $\bar{R} := \mathrm{rank}[\Sigma_\rho]\{\mathrm{rank}[\Sigma_\rho] + \ln(c_2/\delta)\}$ and $d_\mathcal{S}$ is the dimension of the state.

This implies CPPO can learn a policy that can compete against $\pi^*$ with partial coverage $\bar{C}_{\pi^*} < \infty$. Note that the condition $\bar{C}_{\pi^*} < \infty$ does not require $\Sigma_\rho$ to be full-rank. Also the bound uses $\text{rank}[\Sigma_\rho]$ instead of $d$, which means that our bound is distribution dependent and is still valid even when $d = \infty$ as long as the offline data only concentrate on a low-dimensional subspace.

## 5.3 Low-rank MDPs with Representation Learning

We consider the representation learning in offline RL. Following FLAMBE (Agarwal et al., 2020b), we study low-rank MDPs but in the offline setting. Note that low-rank MDPs here are a more generalized model of the aforementioned linear MDPs (Yang & Wang, 2020) since the true feature representation $\phi^\star$ in a low-rank MDP is unknown.

**Definition 4** (Low rank MDPs). *The ground-truth model $P^\star$ admits a low rank decomposition with a dimension $d$ if there exists two embedding functions $\mu^* : \mathcal{S} \rightarrow \mathbb{R}^d, \phi^* : \mathcal{S} \times \mathcal{A} \rightarrow \mathbb{R}^d$ s.t. $P^\star(s' \mid s, a) = \mu^*(s')^\top \phi^*(s, a)$. Neither $\mu^*$ nor $\phi^*$ is known to the learner.*

One interesting special case of a low-rank MDP is the following latent variable model (see Agarwal et al. (2020b) for more details).

**Definition 5** (Latent variable models). *There exists a latent space $\mathcal{Z}$ along with functions $\mu^* : \mathcal{Z} \rightarrow \Delta(\mathcal{S})$ and $\phi^* : \mathcal{S} \times \mathcal{A} \rightarrow \Delta(\mathcal{Z})$ s.t. $P^\star(\cdot \mid s, a) = \sum_{z \in \mathcal{Z}} \mu^*(\cdot \mid z)\phi^*(z \mid s, a)$.*

To tackle representation learning under partial coverage on low-rank MDPs, we setup function classes as follows: given two function classes $\Psi \subset \mathcal{S} \rightarrow \mathbb{R}^d, \Phi \subset \mathcal{S} \times \mathcal{A} \rightarrow \mathbb{R}^d$ (both are realizable in the sense that $\mu^* \in \Psi$ and $\phi^* \in \Phi$), we consider a hypothesis class $\{\mu(s')^\top \phi(s, a); \mu \in \Psi, \phi \in \Phi\}$. Then, CPPO (Algorithm 1) and Theorem 1 still work under this setting. Note that this function class setup is exactly the same as the one from FLAMBE.

Here we show that by leveraging the low-rankness, we can refine the concentrability coefficient to a relative condition number defined by the unknown true representation $\phi^*$. We emphasize that this does not depend on the other features. Particularly, given a comparator policy $\pi^*$, we define $\bar{C}_{\pi^*, \phi^\star}$:

$$\bar{C}_{\pi^*, \phi^\star} = \sup_{x \in \mathbb{R}^d} \frac{x^\top \Sigma_{\pi^*} x}{x^\top \Sigma_\rho x}, \quad \Sigma_{\pi^*} := \mathbb{E}_{s, a \sim d_{P^\star}^{\pi^*}} \phi^*(s, a)\phi^*(s, a)^\top, \quad \Sigma_\rho := \mathbb{E}_{s, a \sim \rho} \phi^*(s, a)\phi^*(s, a)^\top.$$

We can show CPPO learns a policy that can compete against $\pi^*$ as long as $\bar{C}_{\pi^*, \phi^\star} < \infty$.

**Theorem 2** (PAC bound for low-rank MDP). *We set $\xi = c_1 \frac{\ln(|\Phi||\Psi|c_2/\delta)}{n}$. Suppose (a): $\|\phi(s, a)\|_2 \leq 1, \forall(s, a) \in \mathcal{S} \times \mathcal{A}, \forall \phi \in \Phi, \int \mu(s')^\top \phi(s, a)\text{d}(s') = 1$ and $\int \|\mu(s)\|_2 \text{d}s \leq \sqrt{d}, \forall \mu \in \Psi, \phi \in \Phi$, (b) $\rho(s, a) = d_{P^\star}^{\pi_b}(s, a)$, (c) $P^\star(s'|s, a) = \mu^*(s')^\top \phi^*(s, a)$ for some $\mu^* \in \Psi, \phi^* \in \Phi$. With probability at least $1 - \delta$, for all $\pi^* \in \Pi$ (again $\Pi$ can be an unrestricted policy class), CPPO (Algorithm 1) finds $\hat{\pi}$ such that:*

$$V_{P^\star}^{\pi^*} - V_{P^\star}^{\hat{\pi}} \leq c_3 \sqrt{\bar{C}_{\pi^*, \phi^\star} \omega_{\pi^*} \text{rank}(\Sigma_\rho) \frac{\ln(|\Psi||\Phi|c_4/\delta)}{(1-\gamma)^4 n}}, \quad \omega_{\pi^*} = \left(\max_{(s, a)} \frac{\pi^*(a|s)}{\pi_b(a|s)}\right) \quad (3)$$

To the best of our knowledge, this is the first established PAC result under the partial coverage condition $\bar{C}_{\pi^*, \phi^\star} < \infty, \omega_{\pi^*} < \infty$ for low-rank MDPs in the offline setting. We also emphasize that our bound in Theorem 2 is distribution dependent, i.e., it depends on $\text{rank}(\Sigma_\rho)$ rather than the exact rank $d$. Note that $\text{rank}(\Sigma_\rho) \leq d$, and $\text{rank}(\Sigma_\rho)$ could be much smaller than $d$ when the offline distribution only concentrates on a low-dimensional subspace (defined using $\phi^*$). Note that the assumption that $\omega_{\pi^*} < \infty$ does not imply the state-action density ratio $C_{\pi^*, \infty}$ is small. Indeed, $\omega_{\pi^*} < \infty$ is much weaker than $C_{\pi^*, \infty} < \infty$.

## 5.4 Factored MDPs

The last example we include is the factored MDP (Kearns & Koller, 1999) defined as follows:

**Definition 6** (Factored MDPs). *Let $d \in \mathbb{N}^+$ and $\mathcal{O}$ being a small finite set. The state space $\mathcal{S} = \mathcal{O}^d$, and for each state $s$, we denote $s[i] \in \mathcal{O}$ as the $i$-th variable of the state $s$. For each $i \in [1, \cdots, d]$, the parents of $i$, $\text{pa}_i \subset [1, \cdots, d]$, is the subset of state variables that directly influences $i$, i.e., the transition is defined as follows:*

$$\forall s, a, s' : P^\star(s'|s, a) = \prod_{i=1}^d P_i^\star(s'[i]|s[\text{pa}_i], a).$$

*We will denote $\mathcal{S}_i = \mathcal{O}^{|\text{pa}_i|}$, and given $s \in \mathcal{S}$, we will have $s[\text{pa}_i] \in \mathcal{S}_i$*

Due to the factorization, the transition operator $P^\star$ can be described with $L := \sum_{i=1}^d |\mathcal{A}||\mathcal{O}|^{1+|\mathrm{pa}_i|}$ many parameters. In contrast, the non-factored transition will need $O(|\mathcal{O}|^d)$ parameters. When $|\mathrm{pa}_i| \ll d \,\forall i$, it is expected that we can learn this model with lower sample complexity by leveraging the factorization which has been demonstrated in the online setting (Kearns & Koller, 1999). We remark a factored MDP is an example where model-based approaches are necessary as neither the optimal policy nor the Q functions are factored (Koller & Parr, 2000).

We will slightly modify Algorithm 1 to take the factorization into consideration. First, we perform MLE for model learning: each factor $P_i^\star$ is independently learned via MLE:

$$\forall i \in [d], \widehat{P}_{\mathrm{MLE},i} = \arg\max_P \mathbb{E}_{\mathcal{D}}[\ln P(s'[i]|s[\mathrm{pa}_i], a)], \quad \widehat{P} = \prod_i \widehat{P}_{\mathrm{MLE},i}.$$

Next, the constrained policy optimization procedure is defined as

$$\hat{\pi} = \arg\max_\pi \min_{P := \prod_i P_i} V_P^\pi, \text{ s.t.}, \mathbb{E}_{\mathcal{D}}[\mathrm{TV}(P_i(\cdot \mid s, a), \widehat{P}_{\mathrm{MLE},i}(\cdot \mid s, a))^2] \le \xi_i \,(\forall i \in [1, \cdots, d]).$$

Note that in the above objective, there is no restriction on the policy, i.e., the $\arg\max$ operator searches over all possible policies including non-Markovian ones.

To analyze the performance of the above modified CPPO, we introduce a specialized concentration coefficient for factored MDPs that utilizes the factored structure. We focus on density ratio based concentrability coefficients since in a factored MDP with the function class $\mathcal{M} := \{P = \prod_i P_i : P_i \in \mathcal{S}_i \times \mathcal{A} \to \Delta(\mathcal{O})\}$, the concentrability coefficient associated with $\mathcal{M}$ in Definition 1 will be reduced to the density ratio. For any $\pi^*$, we define the concentrability coefficients for the factored MDP as follows:

$$\ddot{C}_{\pi^*,\infty} := \max_{j \in [1,\cdots,d]} \max_{s_j \in \mathcal{S}_j, a \in \mathcal{A}} \frac{d_{P^\star}^{\pi^*}(s_j, a)}{\rho(s_j, a)},$$

where for $s_j \in \mathcal{S}_j$, we denote $\nu(s_j, a) := \sum_{s \in \mathcal{S}: s[\mathrm{pa}_j]=s_j} \nu(s, a)$ for any distribution $\nu \in \Delta(\mathcal{S} \times \mathcal{A})$.

Comparing to $C_{\pi^*,\infty}$ defined on the original state space $\mathcal{S}$, here $\ddot{C}_{\pi^*,\infty}$ is defined over each state space $\mathcal{S}_j$ associated with each factor $j$. Note that when $|\mathrm{pa}_j| = \Theta(1)$, $|\mathcal{S}_j|$ is exponentially smaller than $|\mathcal{S}|$. One can verify that $\ddot{C}_{\pi^*,\infty} \le C_{\pi^*,\infty}$ (see Appendix E.7), where $C_{\pi^*,\infty}$ ignores the factored structure and treat $\mathcal{S}$ as a whole single space. *This formally demonstrates the benefit of the factored structure in terms of the coverage condition in offline RL.*

With the new definition of the concentrability coefficients, now we are ready to state the PAC bound of CPPO for factored MDPs. Recall $L := \sum_{i=1}^d L_i, L_i = |\mathcal{A}||\mathcal{O}|^{1+|\mathrm{pa}_i|}$.

**Theorem 3** (PAC bound for factored MDP). *We set $\xi_i = c_1 \frac{L_i \ln(L_i c_2 d/\delta)}{n}$. Then with probability $1 - \delta$, CPPO finds a policy $\hat{\pi}$ such that for all comparator policy $\pi^* \in \Pi$ ($\Pi$ can be unrestricted),*

$$V_{P^\star}^{\pi^*} - V_{P^\star}^{\hat{\pi}} \le c_3(1 - \gamma)^{-2} \sqrt{\frac{d\ddot{C}_{\pi^*,\infty} L \cdot \ln(nLc_4 d/\delta)}{n}}.$$

Note that our sub-optimality gap scales polynomially with respect to $L$, i.e., the complexity of the factored MDP, rather than $|\mathcal{S}|$ which can be $\Omega(\exp(d))$.

## 6 CONCLUSION

We study model-based offline RL with function approximation under partial coverage. We show that for the model-based setting, realizability in function class and partial coverage together are enough to learn a policy that is comparable to *any* policies (including history-dependent policies) covered by the offline distribution. Our result demonstrates a sharp contrast to model-free offline RL approaches which often require additional structural conditions in the function class (e.g., Bellman completion) and have restrictions on the pool of candidate policies that they can compete against.

Some readers might wonder whether CPPO is computationally efficient. The minimax optimization problem $\arg\max_{\pi \in \Pi} \min_{P \in M} V_P^\pi$ fits into a framework of planning on robust MDPs (Nilim & El Ghaoui, 2005; Iyengar, 2005). By introducing a robust Bellman equation, they proposed value iteration and policy iteration algorithms, and showed that algorithms are practically tractable in the tabular setting. In the non-tabular setting, Lim & Autef (2019); Tamar et al. (2014) propose the extension using function approximation. Thus, we can apply their methods to approximately solve the minimax optimization problem in a model-free fashion. We leave the formal theoretical justification when using these approximation planning algorithms as an important direction for future work.

## ACKNOWLEDGEMENT

The authors would like to thank Nan Jiang, Tengyang Xie for valuable feedback.

Masatoshi Ueharra is partially supported by Masason foundation.

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

## A   GENERALIZATION OF THEOREM 1

We present the generalized version of Theorem 1 when the hypothesis class is infinite. We define
the modified function class of $\mathcal{M}$:

$$\mathcal{H} = \left\{ \sqrt{\frac{P + P^*}{2}} \mid P \in \mathcal{M} \right\}.$$

Given a function class $\mathcal{F}$, let $\mathcal{N}_{[]}(\delta, \mathcal{F}, d)$ be the bracketing number of $\mathcal{F}$ w.r.t the metric $d(a, b)$
given by

$$d(a, b) = \mathbb{E}_{(s,a) \sim \rho} \left[ \int (a(s' \mid s, a) - b(s' \mid s, a))^2 d(s') \right]^{1/2}.$$

Then, the entropy integral of $\mathcal{F}$ is given by

$$J_B(\delta, \mathcal{F}, d) = \max \left( \int_{\delta^2/2}^{\delta} (\log \mathcal{N}_{[]}(u, \mathcal{F}, d))^{1/2} du, \delta \right). \tag{4}$$

We also define the localized class of $\mathcal{H}$:

$$\mathcal{H}(\delta) = \{ h \in \mathcal{H} : \mathbb{E}_{(s,a) \sim \rho}[h^2(P(\cdot \mid s, a) \| P^*(\cdot \mid s, a))] \le \delta^2 \},$$

where $h(P(\cdot \mid s, a) \| P^*(\cdot \mid s, a))$ denotes Hellinger distance defined by

$$\left( 0.5 \int \{ \sqrt{P(s' \mid s, a)} - \sqrt{P^*(s' \mid s, a)} \}^2 d(s') \right)^{1/2}.$$

Based on Theorem 7.4 (van de Geer, 2000), the MLE has the following guarantee.

**Theorem 4** (MLE guarantee with general function approximation). *We take a function $G(\epsilon)$ :
$[0, 1] \to \mathbb{R}$ s.t. $G(\epsilon) \ge J_B[\epsilon, \mathcal{H}(\epsilon), d]$ and $G(\epsilon)/\epsilon^2$ is a non-increasing function w.r.t $\epsilon$. Then,
letting $\xi_n$ be a solution to $\sqrt{n}\epsilon^2 \ge cG(\epsilon)$ w.r.t $\epsilon$. With probability $1 - \delta$, we have*

$$\mathbb{E}_{(s,a) \sim \rho}[\|\hat{P}_{\mathrm{MLE}}(\cdot \mid s, a) - P(\cdot \mid s, a)\|_1^2] \le c_1 \left\{ \xi_n + \sqrt{\log(c_2/\delta)/n} \right\}^2.$$

We remark that the original guarantee in (van de Geer, 2000) is given for the estimation of uncondi-
tional distributions. The adaption to the conditional case is straightforward. For more details, refer
to Section A.1. Besides, when we assume the convexity of the function class, the entropy integral
with bracketing (4) can be replaced with the entropy integral with covering number (Wainwright,
2019, Chapter 14).

By using the above notation and the MLE guarantee, we can generalize Theorem 1.

**Theorem 5** (Finite sample error bound of CPPO with an infinite hypothesis class). *Assume $P^* \in
\mathcal{M}$. Let $f(P)(s, a) = \mathrm{TV}(P(\cdot \mid s, a), P^*(\cdot \mid s, a))^2$. Define*

$$\mathcal{M}_1 = \left\{ P : \mathbb{E}_{(s,a) \sim \rho}[f(P)(s, a)] \le c \left( \xi_n^2 + \frac{\ln(c/\delta)}{n} \right) \right\},$$

$$\mathcal{M}_2 = \left\{ P : \mathbb{E}_{(s,a) \sim \mathcal{D}}[f(P)(s, a)] \le c \left( G(\mathcal{M}_1) + \xi_n^2 + \frac{\ln(c/\delta)}{n} \right) \right\},$$

$$G(\mathcal{M}_1) = \mathbb{E}[\sup_{P \in \mathcal{M}_1} |(\mathbb{E}_{\mathcal{D}} - \mathbb{E}_{\rho})[f(P)]|], \quad G(\mathcal{M}_2) = \mathbb{E}[\sup_{P \in \mathcal{M}_{\mathcal{D}}} |(\mathbb{E}_{\mathcal{D}} - \mathbb{E}_{\rho})[f(P)]|].$$

*Here, in $G(\mathcal{M}_1)$ and $G(\mathcal{M}_2)$, the expectation is taken over the data. We set $\xi = cG(\mathcal{M}_1) + c\xi_n^2 +
c \left( \frac{\ln(c/\delta)}{n} \right)$. Then, for all $\pi^* \in \Pi$, we have*

$$V_{P^*}^{\pi^*} - V_{P^*}^{\hat{\pi}} \le (1 - \gamma)^{-2} c_1 \sqrt{C_{\pi^*}^{\dagger}} \sqrt{G(\mathcal{M}_2) + G(\mathcal{M}_1) + \xi_n^2 + \frac{\ln(c/\delta)}{n}}.$$

This theorem shows once we can calculate $\mathcal{G}_{\mathcal{M}_1}, \mathcal{G}_{\mathcal{M}_2}$ and $\xi_n$, we can obtain the tight rate. Importantly, $\mathcal{G}_{\mathcal{M}_1}$ and $\mathcal{G}_{\mathcal{M}_2}$ are upper-bounded by the localized versions of Rademacher complexities based on symmetrization argument. Hence, their rates are faster than the ones of the nonlocalized versions.

For example, when $|\mathcal{M}|$ is finite, we first have $\xi_n = \sqrt{\ln(|\mathcal{M}|c/\delta)/n}$. Then, from Bernstein's inequality (Wainwright, 2019, Exercise 2.8) and union bound, $G(\mathcal{M}_1)$ is upper-bounded by

$$G(\mathcal{M}_1) \lesssim \underbrace{\xi_n}_{\text{Variance term}} \times \underbrace{\sqrt{\ln(|\mathcal{M}|c/\delta)/n}}_{\text{Union bound}} \lesssim \xi_n^2.$$

Similarly, from empirical Bernstein's inequality,

$$G(\mathcal{M}_2) \lesssim \underbrace{\xi_n}_{\text{Variance term}} \times \underbrace{\sqrt{\ln(|\mathcal{M}|c/\delta)/n}}_{\text{Union bound}} \lesssim \xi_n^2.$$

Then, we can obtain the result of Theorem 1:

$$V_{P^\star}^{\pi^*} - V_{P^\star}^{\hat{\pi}} \leq (1-\gamma)^{-2} c_1 \sqrt{C_{\pi^*}^\dagger} \sqrt{\frac{\ln(|\mathcal{M}|c/\delta)}{n}}, \quad \forall \pi^* \in \Pi.$$

We stress if we use Hoeffeding's inequality above, we would immediately get the slower rate $O(n^{-1/4})$. To calculate $G(\mathcal{M}_1)$ and $G(\mathcal{M}_2)$ in a tight manner, we need to leverage the knowledge that the variance of each element in $\mathcal{M}_1$ and $\mathcal{M}_2$ is controlled from the restriction $P \in \mathcal{M}_1$ or $P \in \mathcal{M}_2$.

## A.1 RATE OF CONVERGENCE OF MAXIMUM LIKELIHOOD ESTIMATION WITH INFINITE HYPOTHESIS CLASS

We aim for obtaining a PAC guarantee of MLE following (van de Geer, 2000). We explain how we should modify the proof of van de Geer (2000) for unconditional density estimation to conditional density estimation. For simplicity, we assume $P^\star > 0$.

We first introduce the notation:

$$\bar{P} = (P + P^\star)/2, g_P = 0.5 \log \frac{\bar{P}}{P^\star}, (\bar{\mathcal{M}})^{1/2} = \left\{ \sqrt{\bar{P}} \mid P \in \mathcal{M} \right\}.$$

Recall

$$h^2(P_1(\cdot \mid s, a), P_2(\cdot \mid s, a)) = \left( 0.5 \int P_1^{1/2}(s' \mid s, a) - P^{1/2}(s' \mid s, a) \mathrm{d}(s') \right)^{0.5}.$$

Here, from Lemma 4.2 (van de Geer, 2000), the following holds:

**Lemma 1** (Some property of Hellinger distance)**.**

$$\mathbb{E}_{(s,a)\sim\rho}[h^2(\bar{P}_1(\cdot \mid s, a), \bar{P}_2(\cdot \mid s, a))] \leq 0.5\mathbb{E}_{(s,a)\sim\rho}[h^2(P_1(\cdot \mid s, a), P_2(\cdot \mid s, a))],$$
$$\mathbb{E}_{(s,a)\sim\rho}[h^2(P(\cdot \mid s, a), P^\star(\cdot \mid s, a))] \leq \mathbb{E}_{(s,a)\sim\rho}[16h^2(\bar{P}(\cdot \mid s, a), P^\star(\cdot \mid s, a))].$$

We also recall Hellinger distance is stronger than TV distance:

**Lemma 2** (Relation of Hellinger distance and TV distance )**.**

$$\mathrm{TV}(P_1(\cdot \mid s, a), P_2(\cdot \mid s, a)) \leq \sqrt{2}h(P_1(\cdot \mid s, a), P_2(\cdot \mid s, a)).$$

The following lemma is useful to connect the log-loss and the Hellinger distance.

**Lemma 3** (Basic Inequality for MLE)**.**

$$\mathbb{E}_{(s,a)\sim\rho}[h^2(\hat{P}_{\mathrm{MLE}}(\cdot \mid s, a), P^\star(\cdot \mid s, a))] \leq (\mathbb{E}_{\mathcal{D}} - \mathbb{E}_{(s,a)\sim\rho, s'\sim P^\star(\cdot|s,a)})[g_P(s, a, s')].$$

This is proved by Lemma 4.1 (van de Geer, 2000). To simplify the notation, we define

$$H^2(P_1, P_2) = \mathbb{E}_{(s,a)\sim\rho}[h^2(P_1(\cdot \mid s, a), P_2(\cdot \mid s, a))].$$

Here, our goal is showing with probability $1 - \delta$,

$$H^2(\hat{P}_{\mathrm{MLE}}, P^\star) \leq \{\xi_n + \sqrt{\log(c_2/\delta)/n}\}^2.$$

This is proved by showing for $x \geq \xi_n$,

$$\mathrm{P}(H^2(\hat{P}_{\mathrm{MLE}}, P^\star) \geq x^2) \leq c \exp(-nx^2/c^2).$$

This corresponds to the statement in Theorem 7.4 (van de Geer, 2000). To prove the above, we first use

$$\mathrm{P}(H^2(\hat{P}_{\mathrm{MLE}}, P^\star) \geq x^2) \leq \mathrm{P}(16H^2(\hat{\bar{P}}_{\mathrm{MLE}}, P^\star) \geq x^2),$$

from Lemma 1. Then, from Lemma 3, this is upper-bounded by

$$\mathrm{P}(\sup_{P \in \mathcal{M}, H^2(\bar{P}, P^\star) \geq x^2/16} \nu_n(g_P) - \sqrt{n}H^2(P, P^\star) \geq 0) \tag{5}$$

where $\nu_n = \sqrt{n}(\mathbb{E}_{\mathcal{D}} - \mathbb{E}_{(s,a) \sim \rho, s' \sim P^\star(\cdot|s,a)})$. To prove the term 5 is less than $c \exp(-n\delta^2/c^2)$, we use Theorem 5.11 (van de Geer, 2000), that is, some uniform inequality based on entropy with bracketing. The rest of the proof is the same as Theorem 7.4 (van de Geer, 2000). In summary, the only difference is we use the distance $H(a, b)$ tailored to the conditional density estimation instead of unconditional density estimation.

## B  MORE DETAILS FOR KNRS

We explain the algorithm and present the PAC guaranteed for KNRs. Here, we denote the dimension of $\mathcal{S}$ by $d_{\mathcal{S}}$.

We tailor Algorithm 1 to KNRs as follows to obtain a tighter guarantee. First, MLE procedure is replaced with $\hat{W}_{\mathrm{MLE}}$ by regularized MLE:

$$\hat{W}_{\mathrm{MLE}} = \underset{W \in \mathbb{R}^{d_{\mathcal{S}} \times d}}{\arg\min} \; \mathbb{E}_{\mathcal{D}}[\|W\phi(s, a) - s'\|_2^2] + \lambda\|W\|_F^2,$$

where $\|\cdot\|_F$ is a Frobenius norm. Then, the final policy optimization procedure is

$$\hat{\pi} = \arg\max_{\pi \in \Pi} \min_{W \in \mathcal{W}_{\mathcal{D}}} V_{P(W)}^\pi, \text{ s.t.}, \mathcal{W}_{\mathcal{D}} = \{W \in \mathbb{R}^{d_{\mathcal{S}} \times d} : \|(\hat{W}_{\mathrm{MLE}} - W)(\Sigma_n)^{1/2}\|_2 \leq \xi\}$$

where $\Sigma_n = \sum_{i=1}^n \phi(s_i, a_i)\phi^\top(s_i, a_i)$. We state the theoretical guarantee for KNRs below.

**Corollary 3** (PAC bound for KNRs). *Assume $\|\phi(s, a)\|_2 \leq 1, \forall(s, a) \in \mathcal{S} \times \mathcal{A}$. We set*

$$\xi = \sqrt{2\lambda\|W^\star\|_2^2 + 8\zeta^2 \left(d_{\mathcal{S}} \ln(5) + \ln(1/\delta) + \bar{\mathcal{I}}_n\right)}, \quad \bar{\mathcal{I}}_n = \ln\left(\det(\Sigma_n)/\det(\lambda\mathbf{I})\right).$$

*Suppose the KNR model is well-specified. By letting $\|W^\star\|_2^2 = O(1), \zeta^2 = O(1), \lambda = O(1)$, with probability $1 - \delta$, for all $\pi^*$, we have*

$$V_{P^\star}^{\pi^*} - V_{P^\star}^{\hat{\pi}} \leq c_1 H^2 \min(d^{1/2}, \bar{R})\sqrt{\bar{R}}\sqrt{\frac{d_{\mathcal{S}}\bar{C}_{\pi^*, P^\star} \ln(1 + n)}{n}}, \quad \text{where } \bar{R} := \mathrm{rank}[\Sigma_\rho]\{\mathrm{rank}[\Sigma_\rho] + \ln(c_2/\delta)\}.$$

The proof is deferred to Section E.5.

## C  MORE RELATED WORKS

We discuss literature related to representation learning in RL.

Representation learning for low-rank MDPs (ground truth feature representation is unknown) in online learning is studied from a model-based perspective (Agarwal et al., 2020b) and model-free perspective (Modi et al., 2021). In the online setting, Zhang et al. (2021a); Papini et al. (2021) also study representation learning under different model assumptions. Comparing with these works, since our setting is offline, the algorithm and analysis are totally different.

In the offline setting, Ni et al. (2021) study dimensionality reduction in a given kernel space, and Hao et al. (2021) study feature selection in sparse linear MDPs. Their focus is different as they do not study PAC guarantees under partial coverage. Ni et al. (2021) assumes the transition operator can be properly embedded into predefined Reproducing Kernel Hilbert Spaces and learns low-dimensional state-action representations via kernelized embedding and low-rank tensor decomposition. However, they did not study the errors for policy optimization after using these learned features. Regarding offline distribution coverage, Ni et al. (2021) assumes that the feature covariance matrix (feature associated with the pre-defined kernel) of the offline distribution is full rank. Hao et al. (2021) studies an OPE problem on sparse linear Bellman complete MDPs in the offline learning setting where they assume all covariance matrices (covariance matrices that correspond to all possible subsets of features) under the offline distribution are full rank as well. We study policy optimization in low-rank MDPs (with unknown feature representation), and we do not assume full coverage, i.e., we do not assume the feature covariance matrix is full rank, and indeed our result is distribution-dependent since it scales with respect to the rank of the covariance matrix that is defined using the ground truth feature representation.

## D    COMPARISON TO XIE ET AL. (2021)

We compare a result in (Xie et al., 2021) to our result in detail. Let $\mathcal{F}$ be a function class for $Q$-functions. Here, we consider a more general version of their algorithm by replacing the original $\mathcal{E}(f, \pi; \mathcal{D})$ in their algorithm with

$$\mathcal{E}(f, \pi; \mathcal{D}) := \mathcal{L}(f, f; \pi, \mathcal{D}) - \min_{g \in \mathcal{G}} \mathcal{L}(g, f; \pi, \mathcal{D}).$$

In their original algorithm, they set $\mathcal{G} = \mathcal{F}$. Here, we consider the version such that a discriminator class $\mathcal{G}$ can be different from $\mathcal{F}$.

They show the PAC result under partial coverage as follows. Here, $\mathcal{T}_{P^\star}^\pi$ is a Bellman operator under $\pi$ and $P^\star$:

$$\mathcal{T}_{P^\star}^\pi : \{\mathcal{S} \times \mathcal{A} \to \mathbb{R}\} \ni f \mapsto r(s, a) + \mathbb{E}_{P^*(s'|s,a)}[f(s', \pi)] \in \{\mathcal{S} \times \mathcal{A} \to \mathbb{R}\}.$$

**Theorem 6** (Extension of Result in (Xie et al., 2021) )**.** *Suppose realizaibility* $Q_{P^\star}^\pi \in \mathcal{F}, \forall \pi \in \Pi$ *and closeness* $\max_{f \in \mathcal{F}} \min_{g \in \mathcal{G}} \mathbb{E}_{s,a \sim \rho}[(g - \mathcal{T}_{P^\star}^\pi f)^2(s, a)] = 0, \forall \pi \in \Pi$. *Then, with* $1 - \delta$, *for any* $\pi^* \in \Pi$, *we have*

$$V_{P^\star}^{\pi^*} - V_{P^\star}^{\hat{\pi}} = O(\sqrt{C^\diamond \ln(|\Pi||\mathcal{F}||\mathcal{G}|/\delta)/n}), \quad C^\diamond = \sup_{f \in \mathcal{F}} \frac{\mathbb{E}_{(s,a) \sim d_{P^\star}^{\pi^*}}[(f - \mathcal{T}f)^2(s, a)]}{\mathbb{E}_{(s,a) \sim \rho}[(f - \mathcal{T}f)^2(s, a)]}.$$

By combining this result with the conversion from model-free results to model-based results in (Chen & Jiang, 2019, Corollary 6), we can obtain the following result under partial coverage.

**Theorem 7** (PAC guarantee from the direct application of (Xie et al., 2021) to mode-based RL )**.** *Assume* $P^\star \in \mathcal{M}$. *Then, there exists an algorithm s.t. with* $1 - \delta$, *for any policy* $\pi^\star \in \Pi$,

$$V_{P^\star}^{\pi^*} - V_{P^\star}^{\hat{\pi}} = O(\sqrt{C^\diamond \ln(|\Pi||\mathcal{M}|/\delta)/n}).$$

As we mentioned, this is worse than our result since it includes $|\Pi|$. Besides, the algorithm can only compete against policies restricted in $\Pi$, while our algorithm works for the unrestricted policy class $\Pi$ which could even include history dependent policies. For completeness, we give the proof as follows.

We remark that their results (Theorem 4.1) with NPG that can possibly compete with any stochastic policies, are not applicable here. This is because they need an assumption that the comparator policy $\pi^*$ needs to satisfy $Q_{P^\star}^{\pi^*} \in \mathcal{F}$ and $\max_{f \in \mathcal{F}} \min_{g \in \mathcal{G}} \mathbb{E}_{s,a \sim \rho}[(g - \mathcal{T}_{P^\star}^{\pi^*} f)^2(s, a)] = 0$, which does not hold for the corresponding Q-function class $\mathcal{F}$ after the conversion. As a notable exception, when the model is a linear Bellman-complete MDP (Zanette et al., 2021), any stochastic policies satisfy the Bellman completeness for the linear Q-function class; then, their algorithms can learn policies that can compete with any stochastic policies satisfying partial coverage.

*Proof of Theorem 7.* Given a model class $\mathcal{M}$, consider the following reduction. We define a $Q$-function class:

$$\mathcal{F} = \{q_P^\pi \mid \pi \in \Pi, P \in \mathcal{M}\}.$$

Then, we define a discriminator class $\mathcal{G}$:

$$\mathcal{G} = \{\mathcal{T}_{P'}^{\pi'} q_P^\pi \mid \pi \in \Pi, \pi' \in \Pi, P \in \mathcal{M}, P' \in \mathcal{M}\}.$$

The above satisfies the realizability $Q_{P^\star}^\pi \in \mathcal{F}, \forall \pi \in \Pi$ and the closedness $\mathcal{T}_{P^\star}^\pi \mathcal{F} \subset \mathcal{G}, \forall \pi \in \Pi$. Thus, the assumptions in Theorem 6 are satisfied. Then, we have

$$V_{P^\star}^{\pi^*} - V_{P^\star}^{\hat\pi} = O(\sqrt{C^\diamond \ln(|\Pi||\mathcal{F}||\mathcal{G}|/\delta)/n})$$
$$= O(\sqrt{C^\diamond \ln(|\Pi||\mathcal{M}|/\delta)/n}),$$

noting $|\mathcal{F}| = |\Pi||\mathcal{M}|$ and $|\mathcal{G}| = |\Pi|^2|\mathcal{M}|^2$.

$\square$

# E    MISSING PROOFS

Below we use $c, c_1, c_2, \cdots$ to denote universal constants. For a $d$-dimensional vector $a$ and a matrix $A \in \mathbb{R}^{d \times d}$, we denote $\|a\|_A^2 = a^\top A a$. Here, $a \lesssim B$ means $a \leq cB$ for some universal constant. $c$

## E.1    PROOFS FOR GENERAL FUNCTION APPROXIMATION

*Proof of Theorem 1.* From Lemma 6, the MLE guarantee gives us the following generalization bound: with probability $1 - \delta$,

$$\mathbb{E}_{s,a\sim\rho}[\text{TV}(\widehat{P}_{\text{MLE}}(\cdot \mid s, a), P^\star(\cdot \mid s, a))^2] \lesssim \frac{\ln(|\mathcal{M}|/\delta)}{n}. \tag{6}$$

Letting

$$A(P) := |\mathbb{E}_{s,a\sim\rho}[\text{TV}(P(\cdot \mid s, a), P^\star(\cdot \mid s, a))^2] - \mathbb{E}_{\mathcal{D}}[\text{TV}(P(\cdot \mid s, a), P^\star(\cdot \mid s, a))^2]|.$$

with probability $1 - \delta$, from union bound and Bernstein's inequality, we also have

$$A(P) \leq \sqrt{\frac{c_1 \text{var}_{(s,a)\sim\rho}[\text{TV}(P(\cdot \mid s, a), P^\star(\cdot \mid s, a))^2]\ln(|\mathcal{M}|/\delta)}{n}} + \frac{c_2 \ln(|\mathcal{M}|/\delta)}{n}, \forall P \in \mathcal{M}. \tag{7}$$

Hereafter, we condition on the above two events. Recall that we construct the version space using $\mathcal{D}$ and $\widehat{P}_{\text{MLE}}$ as follows:

$$\mathcal{M}_{\mathcal{D}} := \left\{ P \in \mathcal{M} : \mathbb{E}_{\mathcal{D}}[\text{TV}(P(\cdot \mid s, a), \hat{P}_{\text{MLE}}(\cdot \mid s, a))^2] \leq \xi \right\}.$$

**First Step: Show $P^\star \in \mathcal{M}_{\mathcal{D}}$ in high-probability.**    We set $\xi = c\frac{\ln(|\mathcal{M}|/\delta)}{n}$. Conditioning on the above two events equations (6) and (7), we prove $P^\star \in \mathcal{M}_{\mathcal{D}}$. This is proved by

$\mathbb{E}_{\mathcal{D}}[\text{TV}(\widehat{P}_{\text{MLE}}(\cdot \mid s, a), P^\star(\cdot \mid s, a))^2]$

$= \mathbb{E}_{\mathcal{D}}[\text{TV}(\widehat{P}_{\text{MLE}}(\cdot \mid s, a), P^\star(\cdot \mid s, a))^2] - \mathbb{E}_{(s,a)\sim\rho}[\text{TV}(\widehat{P}_{\text{MLE}}(\cdot \mid s, a), P^\star(\cdot \mid s, a))^2]$

$+ \mathbb{E}_{(s,a)\sim\rho}[\text{TV}(\widehat{P}_{\text{MLE}}(\cdot \mid s, a), P^\star(\cdot \mid s, a))^2]$

$= \mathbb{E}_{\mathcal{D}}[\text{TV}(\widehat{P}_{\text{MLE}}(\cdot \mid s, a), P^\star(\cdot \mid s, a))^2] - \mathbb{E}_{(s,a)\sim\rho}[\text{TV}(\widehat{P}_{\text{MLE}}(\cdot \mid s, a), P^\star(\cdot \mid s, a))^2] + \frac{c_1 \ln(|\mathcal{M}|/\delta)}{n}$

$\lesssim \sqrt{\frac{\text{var}_{(s,a)\sim\rho}[\text{TV}(\widehat{P}_{\text{MLE}}(\cdot \mid s, a), P^\star(\cdot \mid s, a))^2]\ln(|\mathcal{M}|/\delta)}{n}} + \frac{\ln(|\mathcal{M}|/\delta)}{n}$    (From (7))

$\lesssim \sqrt{\frac{\text{E}_{(s,a)\sim\rho}[\text{TV}(\widehat{P}_{\text{MLE}}(\cdot \mid s, a), P^\star(\cdot \mid s, a))^2]\ln(|\mathcal{M}|/\delta)}{n}} + \frac{\ln(|\mathcal{M}|/\delta)}{n}$

$(\text{TV}(\widehat{P}_{\text{MLE}}(\cdot \mid s, a), P^\star(\cdot \mid s, a))^2 \leq 4)$

$\lesssim \frac{1}{n} \ln(|\mathcal{M}|/\delta).$    (Plug in MLE guarantee)

**Second Step: Show** $\mathbb{E}_{s,a\sim\rho}[\mathrm{TV}(P(\cdot\mid s,a),P^\star(\cdot\mid s,a))^2]\le c\xi,\quad\forall P\in\mathcal{M}_\mathcal{D}$ **in high probability.** We show for any $P\in\mathcal{M}_\mathcal{D}$, the distance between $P^\star$ is sufficiently controlled in terms of TV distance. More concretely (conditioning on the above two events (6) and (7) ), we show

$$\mathbb{E}_{s,a\sim\rho}[\mathrm{TV}(P(\cdot\mid s,a),P^\star(\cdot\mid s,a))^2]\lesssim\xi,\quad\forall P\in\mathcal{M}_\mathcal{D}.$$

In order to observe this, for any $P\in\mathcal{M}_\mathcal{D}$, we have

$$\mathbb{E}_\mathcal{D}[\mathrm{TV}(P(\cdot\mid s,a),P^\star(\cdot\mid s,a))^2]$$
$$\le 2\mathbb{E}_\mathcal{D}[\mathrm{TV}(\widehat{P}_{\mathrm{MLE}}(\cdot\mid s,a),P(\cdot\mid s,a))^2]+2\mathbb{E}_\mathcal{D}[\mathrm{TV}(\widehat{P}_{\mathrm{MLE}}(\cdot\mid s,a),P^\star(\cdot\mid s,a))^2]\le 4\xi$$
$$\text{(From }(a+b)^2\le 2a^2+2b^2.)$$

Thus, we have:

$$\mathbb{E}_{s,a\sim\rho}[\mathrm{TV}(P(\cdot\mid s,a),P^\star(\cdot\mid s,a))^2]$$
$$=\mathbb{E}_{s,a\sim\rho}[\mathrm{TV}(P(\cdot\mid s,a),P^\star(\cdot\mid s,a))^2]-\mathbb{E}_\mathcal{D}[\mathrm{TV}(P(\cdot\mid s,a),P^\star(\cdot\mid s,a))^2]+\mathbb{E}_\mathcal{D}[\mathrm{TV}(P(\cdot\mid s,a),P^\star(\cdot\mid s,a))^2]$$
$$\le A(P)+c\xi. \tag{8}$$

Here, from (7), we have

$$A(P)\le\sqrt{\frac{c_1\mathrm{var}_{(s,a)\sim\rho}[\mathrm{TV}(P(\cdot\mid s,a),P^\star(\cdot\mid s,a))^2]]\ln(|\mathcal{M}|/\delta)}{n}}+\frac{c_2\ln(|\mathcal{M}|/\delta)}{n},\forall P\in\mathcal{M}_\mathcal{D}.$$

Then, for any $P\in\mathcal{M}_\mathcal{D}$, we have

$$A(P)\le\sqrt{\frac{c_1\mathrm{E}_{(s,a)\sim\rho}[\mathrm{TV}(P(\cdot\mid s,a),P^\star(\cdot\mid s,a))^4]\ln(|\mathcal{M}|/\delta)}{n}}+\frac{c_2\ln(|\mathcal{M}|/\delta)}{n}$$
$$\le\sqrt{\frac{4c_1\mathrm{E}_{(s,a)\sim\rho}[\mathrm{TV}(P(\cdot\mid s,a),P^\star(\cdot\mid s,a))^2]\ln(|\mathcal{M}|/\delta)}{n}}+\frac{c_2\ln(|\mathcal{M}|/\delta)}{n}$$
$$([\mathrm{TV}(P(\cdot\mid s,a),P^\star(\cdot\mid s,a))^2]\le 4.)$$
$$\le\sqrt{\frac{4c_1(A(P)+c\xi)\ln(|\mathcal{M}|/\delta)}{n}}+\frac{c_2\ln(|\mathcal{M}|/\delta)}{n}.$$

From $(a+b)^2\le 2a^2+2b^2$,

$$A^2(P)\lesssim\left(\sqrt{\frac{c(A(P)+\xi)\ln(|\mathcal{M}|/\delta)}{n}}+\frac{c\ln(|\mathcal{M}|/\delta)}{n}\right)^2\lesssim\frac{(A(P)+\xi)\ln(|\mathcal{M}|/\delta)}{n}+\left\{\frac{c\ln(|\mathcal{M}|/\delta)}{n}\right\}^2$$
$$\lesssim\frac{(A(P)+\xi)\ln(|\mathcal{M}|/\delta)}{n}\qquad\qquad(\xi\text{ includes }\ln(|\mathcal{M}|/\delta))$$
$$\lesssim\frac{(A(P)+1/n\ln(|\mathcal{M}|/\delta))\ln(|\mathcal{M}|/\delta)}{n}.$$

Then, we have

$$A^2(P)-B_1A(P)-B_2\le 0,\quad B_1=c\ln(|\mathcal{M}|/\delta)/n,\quad B_2=c(1/n)^2\ln(|\mathcal{M}|/\delta)^2.$$

This concludes

$$0\le A(P)\le\frac{B_1+\sqrt{B_1^2+4B_2}}{2}\le c(B_1+\sqrt{B_2})\le c\frac{\ln(|\mathcal{M}|/\delta)}{n}\lesssim\xi.$$

Thus, by using the above $A(P)\lesssim\xi(P\in\mathcal{M}_\mathcal{D})$ and (8), with probability $1-\delta$, we have:

$$\mathbb{E}_{s,a\sim\rho}[\mathrm{TV}(P(\cdot\mid s,a),P^\star(\cdot\mid s,a))^2]\le A(P)+c\xi\lesssim\xi,\quad P\in\mathcal{M}_\mathcal{D}.$$

**Third Step: Calculate the final error bound taking the distribution shift into account** For any $P\in\mathcal{M}_\mathcal{D}$, we prove

$$V_{P^\star}^{\pi^*}-V_P^{\pi^*}\le(1-\gamma)^{-2}c\sqrt{C_{\pi^*}^\dagger}\sqrt{\frac{\ln(|\mathcal{M}|/\delta)}{n}}. \tag{9}$$

For any $P \in \mathcal{M}_{\mathcal{D}}$, this is proved as follows:

$$V_{P^\star}^{\pi^*} - V_P^{\pi^*} \leq (1-\gamma)^{-2} \mathbb{E}_{(s,a) \sim d_{P^\star}^{\pi^*}} [\mathrm{TV}(P(\cdot \mid s,a), P^\star(\cdot \mid s,a))]$$

$$\text{(Simulation lemma, Lemma 5)}$$

$$\leq (1-\gamma)^{-2} \sqrt{\mathbb{E}_{(s,a) \sim d_{P^\star}^{\pi^*}} [\mathrm{TV}(P(\cdot \mid s,a), P^\star(\cdot \mid s,a))^2]}$$

$$\leq (1-\gamma)^{-2} \sqrt{C_{\pi^*}^\dagger \mathbb{E}_{(s,a) \sim \rho} [\mathrm{TV}(P(\cdot \mid s,a), P^\star(\cdot \mid s,a))^2]}$$

$$\leq c(1-\gamma)^{-2} \sqrt{C_{\pi^*}^\dagger} \sqrt{\frac{\ln(|\mathcal{M}|/\delta)}{n}}. \quad \text{(Based on the consequence of the second step)}$$

Combining all things together, with probability $1-\delta$, for any $\pi^* \in \Pi$, we have

$$V_{P^\star}^{\pi^*} - V_{P^\star}^{\hat{\pi}} \leq V_{P^\star}^{\pi^*} - \min_{P \in \mathcal{M}_{\mathcal{D}}} V_P^{\pi^*} + \min_{P \in \mathcal{M}_{\mathcal{D}}} V_P^{\pi^*} - V_{P^\star}^{\hat{\pi}}$$

$$\leq V_{P^\star}^{\pi^*} - \min_{P \in \mathcal{M}_{\mathcal{D}}} V_P^{\pi^*} + \min_{P \in \mathcal{M}_{\mathcal{D}}} V_P^{\hat{\pi}} - V_{P^\star}^{\hat{\pi}} \qquad \text{(definition of } \hat{\pi})$$

$$\leq V_{P^\star}^{\pi^*} - \min_{P \in \mathcal{M}_{\mathcal{D}}} V_P^{\pi^*} \qquad \text{(Fist step, } P^\star \in \mathcal{M}_{\mathcal{D}})$$

$$\lesssim (1-\gamma)^{-2} c_1 \sqrt{C_{\pi^*}^\dagger} \sqrt{\frac{\ln(|\mathcal{M}|c_2/\delta)}{n}}. \qquad \text{(From (9))}$$

$$\square$$

**Remark 2** (To compete with all history-dependent polices). *Consider the case where $\Pi$ is all Markovian polices. We want to show we can compete with all history-dependent non-Markovian polices:*

$$\bar{\Pi} = \left\{ \prod_{i=1}^{\infty} \pi_i \mid \pi_i \in \left[ \left( \prod_{k=1}^{i-1} \mathcal{S} \times \mathcal{A} \right) \rightarrow \Delta(\mathcal{A}) \right] \right\}.$$

*We take an element $\pi^*$ from $\bar{\Pi}$. Then, $V_{P^\star}^{\pi^*}$ and $d_{P^\star}^{\pi^*}$ are still well-defined. Then, every step in the proof still holds. The only step we need to check carefully is this line:*

$$V_{P^\star}^{\pi^*} - V_{P^\star}^{\hat{\pi}} \leq V_{P^\star}^{\pi^*} - \min_{P \in \mathcal{M}_{\mathcal{D}}} V_P^{\pi^*} + \min_{P \in \mathcal{M}_{\mathcal{D}}} V_P^{\pi^*} - V_{P^\star}^{\hat{\pi}}$$

$$\leq V_{P^\star}^{\pi^*} - \min_{P \in \mathcal{M}_{\mathcal{D}}} V_P^{\pi^*} + \min_{P \in \mathcal{M}_{\mathcal{D}}} V_P^{\hat{\pi}} - V_{P^\star}^{\hat{\pi}}.$$

*This is proved by $\max_{\pi \in \bar{\Pi}} V_P^\pi = \max_{\pi \in \Pi} V_P^\pi$ for any $P$.*

### E.2 PROOFS FOR GENERAL FUNCTION APPROXIMATION WITH INFINITE HYPOTHESIS CLASS

*Proof of Theorem 5.* From Theorem 4, the MLE guarantee gives us the following generalization bound: with probability $1-\delta$,

$$\mathbb{E}_{(s,a) \sim \rho} [\mathrm{TV}(\widehat{P}_{\mathrm{MLE}}(\cdot \mid s,a), P^\star(\cdot \mid s,a))^2] \lesssim \left( \xi_n^2 + \frac{\ln(c/\delta)}{n} \right). \qquad (10)$$

We define

$$\mathcal{M}_1 = \left\{ P \in \mathcal{M} : \mathbb{E}_{(s,a) \sim \rho} \left[ \mathrm{TV}(P(\cdot \mid s,a), P^\star(\cdot \mid s,a))^2 \right] \leq c \left( \xi_n^2 + \sqrt{\frac{\ln(c/\delta)}{n}} \right) \right\}.$$

for some large $c$. From a functional Bernstein's inequality (Lemma 12), by defining

$$f(P)(s,a) = \mathrm{TV}(P(\cdot \mid s,a), P^\star(\cdot \mid s,a))^2, \ G(\mathcal{M}_1) = \mathbb{E}[\sup_{P \in \mathcal{M}_1} |(\mathbb{E}_{\mathcal{D}} - \mathbb{E}_\rho)[f(P)]|].$$

with probability $1 - \delta$, we have

$$\sup_{P \in \mathcal{M}_1} |(\mathbb{E}_{\mathcal{D}} - \mathbb{E}_\rho)[f(P)]| \lesssim G(\mathcal{M}_1) + \{G(\mathcal{M}_1) + \sup_{P \in \mathcal{M}_1} \mathbb{E}_\rho[f(P)^2]\}^{1/2} \sqrt{\frac{\log(c_1/\delta)}{n}} + \frac{\log(c_1/\delta)}{n}$$

$$\lesssim G(\mathcal{M}_1) + \{G(\mathcal{M}_1) + \sup_{P \in \mathcal{M}_1} \mathbb{E}_\rho[f(P)]\}^{1/2} \sqrt{\frac{\log(c_1/\delta)}{n}} + \frac{\log(c_1/\delta)}{n}$$

$$\lesssim G(\mathcal{M}_1) + \xi_n^2 + \left(\frac{\ln(c/\delta)}{n}\right). \tag{11}$$

Similarly, by defining

$$\mathcal{M}_2 = \left\{ P : \mathbb{E}_{(s,a)\sim\mathcal{D}}[\mathrm{TV}(P(\cdot \mid s,a), P^\star(\cdot \mid s,a))^2] \le cG(\mathcal{M}_1) + c\xi_n^2 + c\left(\frac{\ln(c/\delta)}{n}\right) \right\},$$

$$G(\mathcal{M}_2) = \mathbb{E}[\sup_{P \in \mathcal{M}_{\mathcal{D}}} |(\mathbb{E}_{\mathcal{D}} - \mathbb{E}_\rho)[f(P)]|],$$

$$Z_n = \sup_{P \in \mathcal{M}_2} \mathbb{E}_{(s,a)\sim\rho}[f(P)(s,a)].$$

from a functional Bernstein's inequality, with probability $1 - \delta$, we have

$$\sup_{P \in \mathcal{M}_2} |(\mathbb{E}_{\mathcal{D}} - \mathbb{E}_\rho)[f(P)]| \lesssim G(\mathcal{M}_2) + \{G(\mathcal{M}_2) + Z_n\}^{1/2} \sqrt{\frac{\log(c_1/\delta)}{n}} + \frac{\log(c_1/\delta)}{n}$$

$$\lesssim G(\mathcal{M}_2) + \sqrt{Z_n \frac{\ln(c/\delta)}{n}} + \frac{\ln(c/\delta)}{n} \tag{12}$$

Hereafter, we condition on the above three events:(10), (11) and (12).

**First step: Show $P^* \in \mathcal{M}_{\mathcal{D}}$ in high probability.** From (10), we have

$$\mathbb{E}_{(s,a)\sim\rho}[f(\hat{P}_{\mathrm{MLE}})(s,a)] \le \xi_n^2 + \frac{\ln(c/\delta)}{n}.$$

Thus, $P^\star \in \mathcal{M}_1$. Then, from (11) and (10),

$$\mathbb{E}_{(s,a)\sim\mathcal{D}}[f(\hat{P}_{\mathrm{MLE}})(s,a)]$$

$$= |\mathbb{E}_{(s,a)\sim\mathcal{D}}[f(\hat{P}_{\mathrm{MLE}})(s,a)] - \mathbb{E}_{(s,a)\sim\rho}[f(\hat{P}_{\mathrm{MLE}})(s,a)]| + \mathbb{E}_{(s,a)\sim\rho}[f(\hat{P}_{\mathrm{MLE}})(s,a)]$$

$$\lesssim G(\mathcal{M}_1) + \xi_n^2 + \frac{\ln(c/\delta)}{n}.$$

Thus, $P^\star \in \mathcal{M}_{\mathcal{D}}$ recalling the definition of $\mathcal{M}_{\mathcal{D}}$ (we set $\xi = G(\mathcal{M}_1) + \xi_n^2 + \frac{\ln(c/\delta)}{n}$).

**Second step: Show the upper bound of $\mathbb{E}_{(s,a)\sim\rho}[f(P)(s,a)]$ for any $P \in \mathcal{M}_{\mathcal{D}}$.** For any $P \in \mathcal{M}_{\mathcal{D}}$, as the proof of Theorem 1, we can prove $P \in \mathcal{M}_2$. Thus,

$$\sup_{P \in \mathcal{M}_{\mathcal{D}}} \mathbb{E}_{(s,a)\sim\rho}[f(P)(s,a)] \le Z_n.$$

Thus, we will hereafter analyze $Z_n$. Here from (12), for any $P \in \mathcal{M}_2$, we have

$$\mathbb{E}_{(s,a)\sim\rho}[f(P)(s,a)] \le \sup_{P \in \mathcal{M}_2} \left( |(\mathbb{E}_{(s,a)\sim\mathcal{D}} - \mathbb{E}_{(s,a)\sim\rho})[f(P)(s,a)]| \right) + \sup_{P \in \mathcal{M}_2} \mathbb{E}_{(s,a)\sim\mathcal{D}}[f(P)(s,a)]$$

$$\lesssim G(\mathcal{M}_2) + \sqrt{Z_n \frac{\ln(c/\delta)}{n}} + G(\mathcal{M}_1) + \xi_n^2 + \frac{\ln(c/\delta)}{n}.$$

Hence,

$$Z_n \le G(\mathcal{M}_2) + \sqrt{Z_n \frac{\ln(c/\delta)}{n}} + G(\mathcal{M}_1) + \xi_n^2 + \frac{\ln(c/\delta)}{n}.$$

This shows

$$Z_n \le G(\mathcal{M}_2) + G(\mathcal{M}_1) + \xi_n^2 + \frac{\ln(c/\delta)}{n}.$$

**Third step: Calculate the final error bound.** Following the proof of Theorem 1, we can prove

$$V_{P^\star}^{\pi^*} - V_{P^\star}^{\hat{\pi}} \le (1-\gamma)^{-2} c_1 \sqrt{C_{\pi^*}^\dagger} \sqrt{G(\mathcal{M}_2) + G(\mathcal{M}_1) + \xi_n^2 + \frac{\ln(c/\delta)}{n}}.$$

$\square$

### E.3 Proofs for Tabular MDPs

*Proof of Corollary 1.* We prove in a similar way as Theorem 1.

**First step** We set $\xi = c \frac{|\mathcal{S}|^2 |\mathcal{A}| \ln(n|\mathcal{S}||\mathcal{A}|c_2/\delta)\}}{n}$. Then, from Lemma 7, with probability $1-\delta$, we can show $P^\star \in \mathcal{M}_\mathcal{D}$ since

$$\mathbb{E}_{s,a\sim\mathcal{D}}\left[\mathrm{TV}(\widehat{P}_{\mathrm{MLE}}(\cdot \mid s,a), P^\star(\cdot \mid s,a))^2\right] \le \xi.$$

Hereafter, we condition on the above event.

**Second step.** Following the second step in the proof of Theorem 1 based on (8), for any $P \in \mathcal{M}_\mathcal{D}$, we have

$$\mathbb{E}_{s,a\sim\rho}\left[\mathrm{TV}(P(\cdot \mid s,a), P^\star(\cdot \mid s,a))^2\right] \le c\xi + A(P) \tag{13}$$

where

$$A(P) := |\mathbb{E}_{s,a\sim\rho}[\mathrm{TV}(P(\cdot \mid s,a), P^\star(\cdot \mid s,a))^2] - \mathbb{E}_\mathcal{D}[\mathrm{TV}(P(\cdot \mid s,a), P^\star(\cdot \mid s,a))^2]|.$$

Our goal here is showing with probability $1-\delta$,

$$A(P) \lesssim \xi, \forall P \in \mathcal{M}_\mathcal{D}. \tag{14}$$

To prove (14), consider an $\epsilon$-net $\{P_1(s,a), \cdots, P_K(s,a)\}$ covering a simplex in terms of $\|\cdot\|_1$ [4] for each fixed pair $(s,a) \in \mathcal{S} \times \mathcal{A}$. We take $\epsilon = 1/n$. Since the covering number $K$ is upper-bounded by $(c/\epsilon)^{|\mathcal{S}|}$ (Wainwright, 2019, Lemma 5.7), we can obtain $\bar{M} = \{P_1, \cdots, P_{K|\mathcal{S}|\times|\mathcal{A}|}\}$ s.t. for any possible $P \subset \mathcal{S} \times \mathcal{A} \to \Delta(\mathcal{S})$, there exists $P_i$ s.t.

$$\mathrm{TV}(P_i(\cdot \mid s,a), P(\cdot \mid s,a)) \le \epsilon, \forall(s,a).$$

This implies for any $P \subset \mathcal{S} \times \mathcal{A} \to \Delta(\mathcal{S})$, there exists $P_i(\cdot \mid s,a)$ s.t. $\forall(s,a)$,

$$\begin{aligned}
&|\mathrm{TV}(P(\cdot \mid s,a), P^\star(\cdot \mid s,a))^2 - \mathrm{TV}(P_i(\cdot \mid s,a), P^\star(\cdot \mid s,a))^2| \\
&\le 4|\mathrm{TV}(P(\cdot \mid s,a), P^\star(\cdot \mid s,a)) - \mathrm{TV}(P_i(\cdot \mid s,a), P^\star(\cdot \mid s,a))| \quad (a^2-b^2=(a-b)(a+b)) \\
&\le 4\mathrm{TV}(P\cdot \mid s,a), P_i(\cdot \mid s,a)) \quad (|\|a\| - \|b\|| \le \|a-b\|) \\
&\le 4\epsilon. \tag{15}
\end{aligned}$$

We often use this property (15) hereafter.

Next, we define $\mathcal{M}' \subset \bar{M}$ so that it covers $\mathcal{M}_\mathcal{D}$. Concretely, we define $\mathcal{M}'$:

$$\mathcal{M}' = \{P \in \bar{M} : \exists P'' \in \mathcal{M}_\mathcal{D}, \mathrm{TV}(P(\cdot \mid s,a), P''(\cdot \mid s,a)) \le \epsilon \quad \forall(s,a)\}. \tag{16}$$

The construction is illustrated in Figure 1. Here, from the definition, for any $P \in \mathcal{M}_\mathcal{D}$, we can also find $P' \in \mathcal{M}'$ s.t.

$$\mathrm{TV}(P(\cdot \mid s,a), P'(\cdot \mid s,a)) \le \epsilon, \forall(s,a).$$

This is because from the definition of $\bar{M}$, we can always find $P \in \bar{\mathcal{M}}$ satisfying the above. Such $P$ belongs to $\mathcal{M}'$ from the definition of $\mathcal{M}'$. We use this fact later.

Then, from (16) and recalling (13), we have

$$\mathbb{E}_{s,a\sim\rho}\left[\mathrm{TV}(P(\cdot \mid s,a), P^\star(\cdot \mid s,a))^2\right] \lesssim \xi + A(P), \quad \forall P \in \mathcal{M}'. \tag{17}$$

---

[4] In the tabular setting, since the state space is countable, it is equivalent to L1 distance.

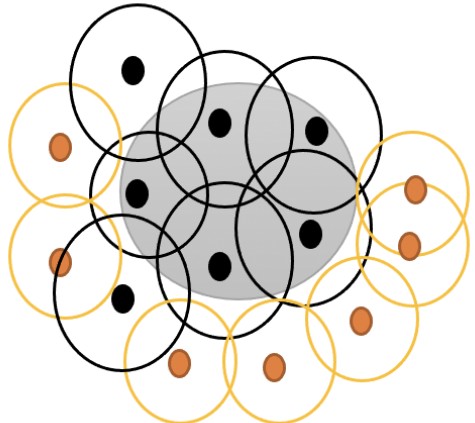

Figure 1: $\mathcal{M}_\mathcal{D}$ is colored in gray. $\mathcal{M}'$ corresponds to the set of black dots. Orange dots correspond to $\bar{\mathcal{M}}$, which do not belong to $\mathcal{M}'$.

because

$$
\mathbb{E}_{s,a\sim\rho}\left[\mathrm{TV}(P(\cdot\mid s,a),P^\star(\cdot\mid s,a))^2\right]
$$
$$
\leq \mathbb{E}_{s,a\sim\rho}\left[\mathrm{TV}(P(\cdot\mid s,a),P''(\cdot\mid s,a))^2\right] + \mathbb{E}_{s,a\sim\rho}\left[\mathrm{TV}(P''(\cdot\mid s,a),P^\star(\cdot\mid s,a))^2\right]
$$
$$
\leq \mathbb{E}_{s,a\sim\rho}\left[\mathrm{TV}(P''(\cdot\mid s,a),P^\star(\cdot\mid s,a))^2\right] + \epsilon^2 \qquad \text{(Take some } P'' \in \mathcal{M}_\mathcal{D} \text{ noting (16))}
$$
$$
\leq c\xi + A(P). \qquad \text{(From (13))}
$$

Then, with probability $1 - \delta$, from Bernstein's inequality, we have

$$
A(P) \leq \sqrt{c\frac{\mathrm{var}[\mathrm{TV}(P(\cdot\mid s,a),P^\star(\cdot\mid s,a))^2]\ln(K^{|\mathcal{S}|\times|\mathcal{A}|}/\delta)}{n}} + \frac{c\ln(K^{|\mathcal{S}|\times|\mathcal{A}|}/\delta)}{n}, \forall P \in \mathcal{M}.
$$

Hereafter, we condition on the above event. Based on (17), we can state

$$
\mathrm{var}[\mathrm{TV}(P(\cdot\mid s,a),P^\star(\cdot\mid s,a))^2] \lesssim \mathrm{E}[\mathrm{TV}(P(\cdot\mid s,a),P^\star(\cdot\mid s,a))^2] \lesssim \xi + A(P), \quad \forall P \in \mathcal{M}',
$$

with probability $1 - \delta$. Following the argument of Theorem 1, for $P \in \mathcal{M}'$, we have

$$
A^2(P) - A(P)B_1 - B_2 \leq 0, \quad B_1 = \frac{\ln(K^{|\mathcal{S}|\times|\mathcal{A}|}/\delta)}{n}, \quad B_2 = \xi\frac{\ln(K^{|\mathcal{S}|\times|\mathcal{A}|}/\delta)}{n} + \left(\frac{\ln(K^{|\mathcal{S}|\times|\mathcal{A}|}/\delta)}{n}\right)^2.
$$

Then, with probability $1 - \delta$, we have

$$
A(P) \leq \frac{\ln(K^{|\mathcal{S}|\times|\mathcal{A}|}/\delta)}{n} + \sqrt{\frac{\ln(K^{|\mathcal{S}|\times|\mathcal{A}|}/\delta)}{n}}\xi^{1/2} \lesssim \xi, \quad \forall P \in \mathcal{M}'. \tag{18}
$$

This shows for any $P \in \mathcal{M}_\mathcal{D}$, we have

$$
|\{\mathbb{E}_\mathcal{D} - \mathbb{E}_{(s,a)\sim\rho}\}[\mathrm{TV}(P(\cdot\mid s,a),P^\star(\cdot\mid s,a))^2]|
$$
$$
\leq |\{\mathbb{E}_\mathcal{D} - \mathbb{E}_{(s,a)\sim\rho}\}[\mathrm{TV}(P'(\cdot\mid s,a),P(\cdot\mid s,a))^2 + \mathrm{TV}(P'(\cdot\mid s,a),P^\star(\cdot\mid s,a))^2]|
$$
$$
\qquad\qquad \text{(We take } P' \in \mathcal{M}' \text{ such that (16))}
$$
$$
\leq |\{\mathbb{E}_\mathcal{D} - \mathbb{E}_{(s,a)\sim\rho}\}[\mathrm{TV}(P'(\cdot\mid s,a),P^\star(\cdot\mid s,a))^2] + 8\epsilon \qquad \text{(From the definition of } \mathcal{M}')
$$
$$
\lesssim \xi. \qquad \text{(From (18) and } P' \in \mathcal{M}')
$$

Thus, (14) is proved.

**Third step.**  We follow the third step of Theorem 1:

$$
V_{P^\star}^{\pi^*} - V_{P^\star}^{\hat{\pi}} \lesssim (1-\gamma)^{-2}\sqrt{C_{\pi^*}^\dagger\xi}.
$$

$\square$

### E.4 PROOFS FOR LINEAR MIXTURE MDPS

*Proof of Corollary 2.* Here, letting $P(\theta) = \theta^\top \psi(s, a, s')$, recall

$$\mathcal{M}_{\text{Mix}} = \left\{ P(\theta) \mid \theta \in \Theta \subset \mathbb{R}^d, \int \theta^\top \psi(s, a, s') \mathrm{d}(s') = 1 \quad \forall (s, a) \right\}, \mathcal{H} = \left\{ \sqrt{\frac{P + P^\star}{2}} \mid P \in \mathcal{M}_{\text{Mix}} \right\}.$$

**Upper-bounding** $\mathbb{E}_{(s,a)\sim\rho}[\text{TV}(P(\theta^\star)(\cdot \mid s, a), P(\hat{\theta}_{\text{MLE}})(\cdot \mid s, a))^2]$.
By invoking Theorem 4, we first show

$$\mathbb{E}_{(s,a)\sim\rho}[\text{TV}(P(\theta^\star)(\cdot \mid s, a), P(\hat{\theta}_{\text{MLE}})(\cdot \mid s, a))^2] \leq c\{(d/n)\ln^2(nR) + \ln(c/\delta)/n\}.$$

To do that, we calculate the entropy integral with bracketing. First, we have
$$\mathcal{N}_{[]}(\epsilon, \mathcal{H}, d) \leq \mathcal{N}_{[]}(\epsilon, \mathcal{M}_{\text{Mix}}, d'). \tag{19}$$

where

$$d'(a, b) = \mathbb{E}_{(s,a)\sim\rho} \left[ \int (a(s, a, s') - b(s, a, s'))^2 \mathrm{d}(s') \right]^{1/2}, \tag{20}$$

$$d(a, b) = \mathbb{E}_{(s,a)\sim\rho} \left[ \int (\sqrt{a(s, a, s')} - \sqrt{b(s, a, s')})^2 \mathrm{d}(s') \right]^{1/2}. \tag{21}$$

Here, we use two observations. The first observation is

$$d^2 \left( \sqrt{\frac{P(\theta') + P^\star}{2}}, \sqrt{\frac{P(\theta'') + P^\star}{2}} \right) \leq c_1 d'^2(P(\theta'), P(\theta''))$$

due to the mean-value theorem
$$\sqrt{a} - \sqrt{b} \leq \max(1/\sqrt{a}, 1/\sqrt{b})(a - b)$$

and assumption $P^\star(s' \mid s, a) \geq c_0 > 0$. The second observation is when we have $P' < g < P''$, we also have $\sqrt{(P' + P^\star)/2} < \sqrt{(g + P^\star)/2} < \sqrt{(P'' + P^\star)/2}$. Then, (19) is concluded.

Next, by letting $\theta^{(1)}, \cdots, \theta^{(K)}$ be an $\epsilon$-cover of the $d$-dimensional ball with a radius $R$, i.e, $B_d(R)$, we have the brackets $\{[P(\theta^{(i)}) - \epsilon, P(\theta^{(i)}) + \epsilon]\}_{i=1}^K$ which cover $\mathcal{M}_{mix}$. This is because for any $P(\theta) \in \mathcal{M}_{mix}$, we can take $\theta^{(i)}$ s.t. $\|\theta - \theta^{(i)}\|_2 \leq \epsilon$, then,
$$P(\theta^{(i)}) - \epsilon < P(\theta) < P(\theta^{(i)}) + \epsilon, \quad \forall (s, a, s')$$

noting
$$|P(\theta)(s, a, s') - P(\theta^{(i)})(s, a, s')| \leq \|\theta - \theta^{(i)}\|_2 \leq \epsilon, \quad \forall (s, a, s') \tag{22}$$
The last equality is from Lemma 10.

The brackets above are size of $\epsilon$. Therefore, we have
$$\mathcal{N}_{[]}(\epsilon, \mathcal{M}_{mix}, \|\cdot\|_2) \leq \mathcal{N}(\epsilon, B_d(cR), \|\cdot\|_2),$$

where $\mathcal{N}(\epsilon, B_d(cR), \|\cdot\|_2)$ is a covering number of $B_d(cR)$ w.r.t $\|\cdot\|_2$. This is upper-bounded by $(cR/\epsilon)^d$ (Wainwright, 2019, Lemma 5.7). Thus, we can calculate the upper bound of the entropy integral $J_B(\delta, \mathcal{M}_{mix}, \|\cdot\|_2)$:

$$\int_0^\delta d^{1/2} \ln^{1/2}(cR/u) \mathrm{d}u \leq \int_0^\delta d^{1/2} \ln(1/u) \mathrm{d}u + \delta d^{1/2} \ln(c_1 R)$$
$$= cd^{1/2}(\delta + \delta \ln(1/\delta)) + \delta d^{1/2} \ln(cR)$$
$$\leq cd^{1/2}\delta \ln(cR/\delta).$$

By taking $G(x) = d^{1/2}x \ln(cR/x)$ in Theorem 4, $\delta_n = (d/n)^{1/2} \ln(nR)$ satisfies the critical inequality
$$\sqrt{n}\delta_n^2 \geq d^{1/2}\delta_n \ln(cR/\delta_n).$$
Finally, with probability $1 - \delta$

$$\mathbb{E}_{(s,a)\sim\rho}[\text{TV}(P(\theta^\star)(\cdot \mid s, a), P(\hat{\theta}_{\text{MLE}})(\cdot \mid s, a))^2] \leq \xi', \quad \xi' := \{(d/n)\ln^2(nR) + \ln(c/\delta)/n\}. \tag{23}$$

Hereafter, we condition on this event.

**Upper bounding** $\mathbb{E}_{\mathcal{D}}[\mathrm{TV}(P(\theta^\star)(\cdot \mid s,a), P(\hat{\theta}_{\mathrm{MLE}})(\cdot \mid s,a))^2]$. We take an $\epsilon$-cover of the ball $B_d(R)$ in terms of $\|\cdot\|_2$, i.e., $\bar{M} = \{\theta^{(1)}, \cdots, \theta^{(K)}\}$, where $K = (cR/\epsilon)^d$. We take $\epsilon = 1/n$. Then, for any $\theta \in B_d(R)$, there exists $\theta^{(i)}$ s.t. $\forall (s,a)$,

$$|\mathrm{TV}(P(\theta)(\cdot \mid s,a), P(\hat{\theta}_{\mathrm{MLE}})(\cdot \mid s,a))^2 - \mathrm{TV}(P(\theta^{(i)})(\cdot \mid s,a), P(\hat{\theta}_{\mathrm{MLE}})(\cdot \mid s,a))^2|$$

$$\leq 4|\mathrm{TV}(P(\theta)(\cdot \mid s,a), P(\hat{\theta}_{\mathrm{MLE}})(\cdot \mid s,a)) - \mathrm{TV}(P(\theta^{(i)})(\cdot \mid s,a), P(\hat{\theta}_{\mathrm{MLE}})(\cdot \mid s,a))|$$
$$(a^2 - b^2 = (a-b)(a+b))$$

$$\leq 4\mathrm{TV}(P(\theta)(\cdot \mid s,a), P(\theta^{(i)})(\cdot \mid s,a)) \qquad\qquad (|\|a\| - \|b\|| \leq \|a-b\|)$$

$$\leq 4\|\theta - \theta^{(i)}\|_2 \qquad\qquad\qquad\qquad\qquad\qquad\text{(From Lemma 10.)}$$

$$\leq 4\epsilon. \qquad\qquad\qquad\qquad\qquad\qquad\qquad\qquad\qquad (24)$$

Hereafter, we condition on the event:

$$|(\mathbb{E}_{\mathcal{D}} - \mathbb{E}_{(s,a)\sim\rho})[\mathrm{TV}(P(\theta)(\cdot \mid s,a), P(\theta^\star)(\cdot \mid s,a))^2]| \qquad\qquad (25)$$

$$\lesssim \sqrt{\frac{\mathrm{var}_{(s,a)\sim\rho}[\mathrm{TV}(P(\theta)(\cdot \mid s,a), P(\theta^\star)(\cdot \mid s,a))^2] \ln(K/\delta)}{n}} + \frac{\ln(K/\delta)}{n}, \quad \forall \theta \in \bar{M}.$$

This event holds with probability $1 - \delta$ from Bernstein's inequality.

Here, note for $\hat{\theta}_{\mathrm{MLE}}$, we have $\theta^{(i)}$ s.t. $\|\hat{\theta}_{\mathrm{MLE}} - \theta^{(i)}\|_2 \leq \epsilon$. Then, following the first step of Theorem 1,

$$\mathbb{E}_{(s,a)\sim\mathcal{D}}\mathrm{TV}(P(\theta^\star)(\cdot \mid s,a), P(\hat{\theta}_{\mathrm{MLE}})(\cdot \mid s,a))^2$$

$$\lesssim \mathbb{E}_{(s,a)\sim\mathcal{D}}\mathrm{TV}(P(\theta^\star)(\cdot \mid s,a), P(\theta^{(i)})(\cdot \mid s,a))^2 + \epsilon \qquad\qquad\text{(From (24))}$$

$$\lesssim (\mathbb{E}_{\mathcal{D}} - \mathbb{E}_{(s,a)\sim\rho})\mathrm{TV}(P(\theta^\star)(\cdot \mid s,a), P(\theta^{(i)})(\cdot \mid s,a))^2 + \epsilon + \mathbb{E}_{(s,a)\sim\rho}\mathrm{TV}(P(\theta^\star)(\cdot \mid s,a), P(\theta^{(i)})(\cdot \mid s,a))^2$$

$$\lesssim \sqrt{\frac{\mathrm{var}_{(s,a)\sim\rho}[\mathrm{TV}(P(\theta^\star)(\cdot \mid s,a), P(\theta^{(i)})(\cdot \mid s,a))^2] \ln(K/\delta)}{n}} + \frac{\ln(K/\delta)}{n}$$

$$+ \epsilon + \mathbb{E}_{(s,a)\sim\rho}\mathrm{TV}(P(\theta^\star)(\cdot \mid s,a), P(\theta^{(i)})(\cdot \mid s,a))^2 \qquad\qquad\text{(From (25))}$$

$$\lesssim \sqrt{\frac{\mathbb{E}_{(s,a)\sim\rho}[\mathrm{TV}(P(\theta^\star)(\cdot \mid s,a), P(\theta^{(i)})(\cdot \mid s,a))^2] \ln(K/\delta)}{n}} + \frac{\ln(K/\delta)}{n}$$

$$+ \epsilon + \mathbb{E}_{(s,a)\sim\rho}[\mathrm{TV}(P(\theta^\star)(\cdot \mid s,a), P(\theta^{(i)})(\cdot \mid s,a))^2].$$
$$(\mathrm{TV}(P(\theta^\star)(\cdot \mid s,a), P(\theta^{(i)})(\cdot \mid s,a))^2 \leq 4)$$

Then, we have

$$\mathbb{E}_{\mathcal{D}}[\mathrm{TV}(P(\theta^\star)(\cdot \mid s,a), P(\hat{\theta}_{\mathrm{MLE}})(\cdot \mid s,a))^2]$$

$$\lesssim \sqrt{\frac{\{\mathbb{E}_{(s,a)\sim\rho}[\mathrm{TV}(P(\theta^\star)(\cdot \mid s,a), P(\hat{\theta}_{\mathrm{MLE}})(\cdot \mid s,a))^2] + \epsilon\} \ln(K/\delta)}{n}} + \frac{\ln(K/\delta)}{n}$$

$$+ \epsilon + \mathbb{E}_{(s,a)\sim\rho}\mathrm{TV}(P(\theta^\star)(\cdot \mid s,a), P(\hat{\theta}_{\mathrm{MLE}})(\cdot \mid s,a))^2$$

$$\lesssim \sqrt{\frac{\{\xi' + \epsilon\} \ln(K/\delta)}{n}} + \frac{\ln(K/\delta)}{n} + \epsilon + \xi'. \qquad\qquad\text{(From (23))}$$

In the end, by taking $\epsilon = 1/n$, we have with probability $1 - \delta$,

$$\mathbb{E}_{\mathcal{D}}\mathrm{TV}(P(\theta^\star)(\cdot \mid s,a), P(\hat{\theta}_{\mathrm{MLE}})(\cdot \mid s,a))^2 \leq \xi, \quad \xi = c\{(d/n)\ln^2(nR) + \ln(c/\delta)/n\}.$$

This also implies with probability $1 - \delta$, $P^\star \in \mathcal{M}_{\mathcal{D}}$.

**Show** $\mathbb{E}_{(s,a)\sim\rho}\mathrm{TV}(P(\theta^\star)(\cdot \mid s,a), P(\theta)(\cdot \mid s,a))^2 \lesssim \xi, \forall P(\theta) \in \mathcal{M}_{\mathcal{D}}$.
We show for any $P \in \mathcal{M}_{\mathcal{D}}$, the distance between $P^\star$ is controlled in terms of TV distance. Our goal is showing

$$\mathbb{E}_{(s,a)\sim\rho}\mathrm{TV}(P(\theta^\star)(\cdot \mid s,a), P(\theta)(\cdot \mid s,a))^2 \lesssim \xi, \quad \forall P(\theta) \in \mathcal{M}_{\mathcal{D}}.$$

First, following the second step of Theorem 1 based on equation 8, we have

$$\mathbb{E}_{(s,a)\sim\rho}\mathrm{TV}(P(\theta^\star)(\cdot \mid s,a), P(\theta)(\cdot \mid s,a))^2 \le A(\theta) + c\xi, \quad \forall P(\theta) \in \mathcal{M}_\mathcal{D} \tag{26}$$

where

$$A(\theta) \coloneqq |(\mathbb{E}_\mathcal{D} - \mathbb{E}_{(s,a)\sim\rho})\mathrm{TV}(P(\theta^\star)(\cdot \mid s,a), P(\theta)(\cdot \mid s,a))^2|.$$

From now on, we again consider an $\epsilon$-cover of the ball $B_d(R)$ in terms of $\| \cdot \|_2$, i.e., $\bar{M} = \{\theta^{(1)}, \cdots, \theta^{(K)}\}$, where $K = (c_1 R/\epsilon)^d$ ($\epsilon = 1/n$). This also covers the space $\mathcal{M}_\mathcal{D}$. We take $\mathcal{M}' = \{\theta^{(i_1)}, \theta^{(i_2)} \cdots, \} \subset \mathcal{M}$ which covers $\mathcal{M}_\mathcal{D}$, that is,

$$\mathcal{M}' = \left\{\theta \in \bar{M} \mid \exists\theta' \mathrm{s.t.} P(\theta') \in \mathcal{M}_\mathcal{D}, \|\theta - \theta'\|_2 \le \epsilon\right\}.$$

Recall Figure 1, which illustrates this definition. Here, for any $\theta$ s.t. $\forall P(\theta) \in \mathcal{M}_\mathcal{D}$, we can take $\theta' \in \mathcal{M}'$ s.t. $\|\theta - \theta'\|_2 \le 1/n$. This is because we can take $\theta \in \bar{M}$ s.t. $\|\theta - \theta'\|_2 \le \epsilon$ noting $\bar{M}$ is an $\epsilon$-net, but such $\theta$ belongs to $\mathcal{M}'$ from the definition of $\mathcal{M}'$.

Then, we have

$$\mathbb{E}_{(s,a)\sim\rho}\mathrm{TV}(P(\theta^\star)(\cdot \mid s,a), P(\theta)(\cdot \mid s,a))^2 \le A(\theta) + c\xi, \quad \forall\theta \in \mathcal{M}'. \tag{27}$$

This is because for any $\theta^{(i)} \in \mathcal{M}'$, we can take $P(\theta) \in \mathcal{M}_\mathcal{D}$ such that

$$\mathbb{E}_{(s,a)\sim\rho}\mathrm{TV}(P(\theta^\star)(\cdot \mid s,a), P(\theta^{(i)})(\cdot \mid s,a))^2$$
$$\le \mathbb{E}_{(s,a)\sim\rho}[\mathrm{TV}(P(\theta^\star)(\cdot \mid s,a), P(\theta^{(i)})(\cdot \mid s,a))^2 - \mathrm{TV}(P(\theta^\star)(\cdot \mid s,a), P(\theta)(\cdot \mid s,a))^2]$$
$$+ \mathbb{E}_{(s,a)\sim\rho}[\mathrm{TV}(P(\theta^\star)(\cdot \mid s,a), P(\theta)(\cdot \mid s,a))^2]$$
$$\le 4\epsilon + \mathbb{E}_{(s,a)\sim\rho}[\mathrm{TV}(P(\theta^\star)(\cdot \mid s,a), P(\theta)(\cdot \mid s,a))^2] \qquad (\|\theta - \theta^{(i)}\|_2 \le \epsilon \text{ and from (24)})$$
$$\lesssim A(\theta) + \xi. \qquad (\text{ From (26)})$$

Here, from (25), we have

$$A(\theta) \le \sqrt{c\frac{\mathrm{var}_{(s,a)\sim\rho}[\mathrm{TV}(P(\theta^\star)(\cdot \mid s,a), P(\theta)(\cdot \mid s,a))^2]\ln(K/\delta)}{n}} + \frac{c\ln(K/\delta)}{n}, \quad \forall\theta \in \mathcal{M}'$$

Based on the construction of $\mathcal{M}'$ and (27), we have

$$\mathrm{var}_{(s,a)\sim\rho}[\mathrm{TV}(P(\theta^\star)(\cdot \mid s,a), P(\theta)(\cdot \mid s,a))^2] \lesssim A(\theta) + \xi, \quad \forall\theta \in \mathcal{M}'.$$

Then, following the second step of Theorem 1, $A(\theta)$ satisfies

$$A^2(\theta) - A(\theta)B_1 - B_2 \le 0, \quad B_1 = \frac{\ln(K/\delta)}{n}, B_2 = \xi\frac{\ln(K/\delta)}{n} + \left(\frac{\ln(K/\delta)}{n}\right)^2.$$

Then, we have

$$A(\theta) \le \frac{\ln(K/\delta)}{n} + \xi^{1/2}\sqrt{\frac{\ln(K/\delta)}{n}} \lesssim \xi, \quad \forall\theta \in \mathcal{M}'. \tag{28}$$

We combine all steps. Recall for any $\forall P(\theta) \in \mathcal{M}_\mathcal{D}$, we can take $\theta' \in \mathcal{M}'$ s.t. $\|\theta - \theta'\|_2 \le 1/n$. Then, for any $P(\theta) \in \mathcal{M}_\mathcal{D}$, we have

$$A(\theta) = |(\mathbb{E}_\mathcal{D} - \mathbb{E}_{(s,a)\sim\rho})\mathrm{TV}(P(\theta)(\cdot \mid s,a), P(\theta^\star)(\cdot \mid s,a))^2$$
$$\le |(\mathbb{E}_\mathcal{D} - \mathbb{E}_{(s,a)\sim\rho})[\mathrm{TV}(P(\theta)(\cdot \mid s,a), P(\theta^\star)(\cdot \mid s,a))^2 - \mathrm{TV}(P(\theta')(\cdot \mid s,a), P(\theta^\star)(\cdot \mid s,a))^2]$$
$$+ (\mathbb{E}_\mathcal{D} - \mathbb{E}_{(s,a)\sim\rho})[\mathrm{TV}(P(\theta')(\cdot \mid s,a), P(\theta^\star)(\cdot \mid s,a))^2]$$
$$\le 8\epsilon + |(\mathbb{E}_\mathcal{D} - \mathbb{E}_{(s,a)\sim\rho})[\mathrm{TV}(P(\theta')(\cdot \mid s,a), P(\theta^\star)(\cdot \mid s,a))^2] \qquad (\text{From equation 24})$$
$$\lesssim \xi. \qquad (\text{From equation 28 and } \theta' \in \mathcal{M}')$$

Then, we have with probability $1 - \delta$,

$$A(\theta) \lesssim \xi, \quad \forall P(\theta) \in \mathcal{M}_\mathcal{D}. \tag{29}$$

Finally, for any $P(\theta) \in \mathcal{M}_\mathcal{D}$, with probability $1 - \delta$, we have

$$\mathbb{E}_{(s,a)\sim\rho}[\mathrm{TV}(P(\theta^\star)(\cdot \mid s,a), P(\theta)(\cdot \mid s,a))^2] \le A(\theta) + c\xi \qquad (\text{From equation 26})$$
$$\lesssim \xi. \qquad (\text{From equation 29})$$

**Distribution shift part**    Here, for $P \in \mathcal{M}_\mathcal{D}$ we prove

$$V_{P^\star}^{\pi^*} - V_P^{\pi^*} \lesssim (1-\gamma)^{-2}\sqrt{dC_{\pi^*,\mathrm{mix}}\xi}, \tag{30}$$

$$V_{P^\star}^{\pi^*} - V_P^{\pi^*} \lesssim (1-\gamma)^{-2}\sqrt{C_{\pi^*}^\dagger\xi}. \tag{31}$$

Following the third step of the proof of Theorem 5, this immediately concludes the bound

$$V_{P^\star}^{\pi^*} - V_{P^\star}^{\hat{\pi}} \lesssim (1-\gamma)^{-2}\sqrt{dC_{\pi^*,\mathrm{mix}}\xi},$$

$$V_{P^\star}^{\pi^*} - V_{P^\star}^{\hat{\pi}} \lesssim (1-\gamma)^{-2}\sqrt{C_{\pi^*}^\dagger\xi}.$$

Since (31) is obvious from simulation lemma, we only prove (30). To prove (30), we take a distribution $P(\theta) \in \mathcal{M}_\mathcal{D}$. First, recall for $P(\theta) \in \mathcal{M}_\mathcal{D}$, we have

$$\mathbb{E}_{(s,a)\sim\rho}\mathrm{TV}(P(\theta^\star)(\cdot \mid s,a), P(\theta)(\cdot \mid s,a))^2 \lesssim \xi.$$

From the third statement of Lemma 10, for any $V : \mathcal{S} \to [0,1]$, we have

$$\mathbb{E}_{(s,a)\sim\rho}[|(\theta - \theta^*)^\top\psi_V(s,a)|^2] \lesssim \xi.$$

Thus,

$$\forall V : \mathcal{S} \to [0,1], \quad (\theta - \theta^*)^\top\Sigma_{\rho,V}(\theta - \theta^*) \lesssim \xi, \quad \Sigma_{\rho,V} = \mathbb{E}_{(s,a)\sim\rho}[\psi_V(s,a)\psi_V^\top(s,a)].$$

Here, we have

$$V_{P^\star}^{\pi^*} - V_P^{\pi^*} \le (1-\gamma)^{-1}\left|\mathbb{E}_{(s,a)\sim d_{P^\star}^{\pi^*}}\left[\int\{P(s' \mid s,a) - P^\star(s' \mid s,a)\}V_P^{\pi^*}(s')\mathrm{d}(s')\right]\right|$$

$$\text{(Simulation lemma, Lemma 5)}$$

$$\le (1-\gamma)^{-1}\left|\mathbb{E}_{(s,a)\sim d_{P^\star}^{\pi^*}}\left[(\theta - \theta^*)\psi_{V_P^{\pi^*}}(s,a)\right]\right|$$

$$\text{(Recall } \psi_V = \int\psi(s,a,s')V_P^{\pi^*}(s')\mathrm{d}(s'))$$

$$\le (1-\gamma)^{-1}\underbrace{\|\theta - \theta^*\|_{\lambda I+\Sigma_{\rho,V_P^{\pi^*}}}}_{(a)}\underbrace{\mathbb{E}_{(s,a)\sim d_{P^\star}^{\pi^*}}\left[\|\psi_{V_P^{\pi^*}}(s,a)\|_{(\Sigma_{\rho,V_P^{\pi^*}}+\lambda I)^{-1}}\right]}_{(b)}.$$

$$\text{(CS inequality)}$$

The first term (a) is upper-bounded by $\sqrt{\{(1-\gamma)^{-2}\xi + \lambda R^2\}}$ noting $0 \le V_P^{\pi^*} \le (1-\gamma)^{-1}$. The term (b) is upper-bounded by

$$\mathbb{E}_{(s,a)\sim d_{P^\star}^{\pi^*}}\left[\|\psi_{V_P^{\pi^*}}(s,a)\|_{(\Sigma_{\rho,V_P^{\pi^*}}+\lambda I)^{-1}}\right] \le \mathbb{E}_{(s,a)\sim d_{P^\star}^{\pi^*}}\left[\|\psi_{V_P^{\pi^*}}(s,a)\|_{(\Sigma_{\rho,V_P^{\pi^*}}+\lambda I)^{-1}}^2\right]^{1/2}$$

$$\text{(Jensen's inequality)}$$

$$= \sqrt{\mathrm{Tr}(\Sigma_{d_{P^\star}^{\pi^*},V_P^{\pi^*}}(\lambda I + \Sigma_{\rho,V_P^{\pi^*}})^{-1})}$$

$$\le \sqrt{C_{\pi^*,\mathrm{mix}}\mathrm{Tr}(\Sigma_{\rho,V_P^{\pi^*}}(\lambda I + \Sigma_{\rho,V_P^{\pi^*}})^{-1})}$$

$$\text{(From Lemma 11)}$$

$$\le \sqrt{C_{\pi^*,\mathrm{mix}}\mathrm{rank}(\Sigma_{\rho,V_P^{\pi^*}})} \le \sqrt{C_{\pi^*,\mathrm{mix}}d}.$$

By taking $\lambda$ s.t. $\lambda R^2 \lesssim (1-\gamma)^{-2}\xi$, (30) is proved.

For linear MDPs, from the fourth statement of Lemma 10, $C_{\pi^*,\mathrm{mix}} \le \bar{C}_{\pi^*}$. Then, the statement is concluded. □

### E.5    PROOFS FOR KNRS

*Proof of Corollary 3.*    We prove in a similar way as Theorem 1.

**First Step**  Recall

$$\xi = \sqrt{2\lambda\|W^\star\|_2^2 + 8\zeta^2\left(d_{\mathcal{S}}\ln(5) + \ln(1/\delta) + \bar{\mathcal{I}}_n\right)}, \quad \bar{\mathcal{I}}_n = \ln\left(\det(\Sigma_n)/\det(\lambda\mathbf{I})\right).$$

Thus, from Lemma 8, with probability $1 - \delta$, we can show $W^* \in \mathcal{W}_{\mathcal{D}}$ since

$$\left\|\left(\widehat{W}_{\mathrm{MLE}} - W^\star\right)(\Sigma_n)^{1/2}\right\|_2 \leq \xi.$$

Hereafter, we condition on this event.

**Second step**  For any $W \in \mathcal{W}_{\mathcal{D}}$, with probability $1 - \delta$, we have

$$\left\|(W - W^\star)(\Sigma_n)^{1/2}\right\|_2 \leq \left\|\left(W - \widehat{W}\right)(\Sigma_n)^{1/2}\right\|_2 + \left\|\left(W^* - \widehat{W}\right)(\Sigma_n)^{1/2}\right\|_2 \leq \xi.$$

**Third step**  Note $P^\star = P(W^*)$. Then,

$$\begin{aligned}
V_{P^\star}^{\pi^*} - V_{P^\star}^{\hat{\pi}} &\leq V_{P^\star}^{\pi^*} - \min_{W \in \mathcal{W}_{\mathcal{D}}} V_{P(W)}^{\pi^*} + \min_{W \in \mathcal{W}_{\mathcal{D}}} V_{P(W)}^{\pi^*} - V_{P^\star}^{\hat{\pi}} \\
&\leq V_{P^\star}^{\pi^*} - \min_{W \in \mathcal{W}_{\mathcal{D}}} V_{P(W)}^{\pi^*} + \min_{W \in \mathcal{W}_{\mathcal{D}}} V_{P(W)}^{\hat{\pi}} - V_{P^\star}^{\hat{\pi}} \qquad &\text{(definition of } \hat{\pi}) \\
&\leq V_{P^\star}^{\pi^*} - \min_{W \in \mathcal{W}_{\mathcal{D}}} V_{P(W)}^{\pi^*}. \qquad &\text{(Fist step, } W^* \in \mathcal{W}_{\mathcal{D}})
\end{aligned}$$

Then, by setting $W' = \arg\min_{W \in \mathcal{M}_{\mathcal{D}}} V_{P(W)}^{\pi^*}$, we have

$$\begin{aligned}
V_{P^\star}^{\pi^*} - V_{P^\star}^{\hat{\pi}} &\leq (1-\gamma)^{-2}\mathbb{E}_{(s,a)\sim d_{P^\star}^{\pi^*}}[\|P'(s,a) - P^\star(s,a)\|_{\mathrm{TV}}] \\
&\leq \frac{(1-\gamma)^{-2}}{\zeta}\mathbb{E}_{(s,a)\sim d_{P^\star}^{\pi^*}}[\|(W' - W^\star)\phi(s,a)\|_2] \qquad &\text{(Lemma 9)} \\
&\leq \frac{(1-\gamma)^{-2}}{\zeta}\mathbb{E}_{(s,a)\sim d_{P^\star}^{\pi^*}}\left[\left\|(W' - W^\star)(\Sigma_n)^{1/2}\right\|_2 \|\phi(s,a)\|_{\Sigma_n^{-1}}\right] \qquad &\text{(CS inequality)} \\
&\leq \frac{(1-\gamma)^{-2}}{\zeta}\xi\mathbb{E}_{(s,a)\sim d_{P^\star}^{\pi^*}}[\|\phi(s,a)\|_{\Sigma_n^{-1}}] \qquad &\text{(Second step)}
\end{aligned}$$

From Chang et al. (2021, Theorem 20), with probability $1 - \delta$, we have

$$\xi \leq c_1\sqrt{\|W^*\|_2 + d_{\mathcal{S}}\min(\mathrm{rank}(\Sigma_\rho)\{\mathrm{rank}(\Sigma_\rho) + \ln(c_2/\delta)\}, d)\ln(1+n)}.$$

In addition, from Chang et al. (2021, Theorem 21), with probability $1 - \delta$, we also have

$$\mathbb{E}_{(s,a)\sim d_{P^\star}^{\pi^*}}[\|\phi(s,a)\|_{\Sigma_n^{-1}}] \leq c_1\sqrt{\frac{\bar{C}_{\pi^*}\mathrm{rank}[\Sigma_\rho]\{\mathrm{rank}[\Sigma_\rho] + \ln(c_2/\delta)\}}{n}}.$$

Finally, by combining all things, we have

$$V_{P^\star}^{\pi^*} - V_{P^\star}^{\hat{\pi}} \leq c_1(1-\gamma)^{-2}\min(d^{1/2}, \bar{R})\sqrt{\bar{R}}\sqrt{\frac{d_{\mathcal{S}}\bar{C}_{\pi^*}\ln(1+n)}{n}}, \bar{R} = \mathrm{rank}[\Sigma_\rho]\{\mathrm{rank}[\Sigma_\rho] + \ln(c_2/\delta)\}.$$

$$\square$$

### E.6  PROOFS FOR LOW-RANK MDPS

*Proof of Theorem 2.*  Until the second step, we can perform the same analysis as Theorem 1. More concretely, with probability $1 - \delta$, we have $P^\star \in \mathcal{M}_{\mathcal{D}}$ and

$$\mathbb{E}_{s,a\sim\rho}[\mathrm{TV}(P(\cdot \mid s,a), P^\star(\cdot \mid s,a))^2] \leq \xi, \quad \forall P \in \mathcal{M}_{\mathcal{D}}, \xi := c\frac{\ln(|\mathcal{M}|/\delta)}{n}. \qquad (32)$$

Hereafter, we condition on the above event.

Letting $f(s,a) = \mathrm{TV}(P(\cdot \mid s,a), P^\star(\cdot \mid s,a))$, we use Lemma 4. Then,

$$\mathbb{E}_{(s,a)\sim d_{P^\star}^\pi}[f(s,a)] \le \mathbb{E}_{(s,a)\sim d_{P^\star}^\pi}[\|\phi^\star(s,a)\|_{\Sigma_{\rho,\phi^\star}^{-1}}]\sqrt{n\gamma\omega_\pi\mathbb{E}_\rho[f^2(s,a)] + 4\gamma^2\lambda d} + \sqrt{(1-\gamma)\omega_\pi\mathbb{E}_\rho[f^2(s,a)]}$$

where $\Sigma = n\mathbb{E}_\rho[\phi^\star\phi^{\star\top}] + \lambda I$. We consider how to bound $\mathbb{E}_{(s,a)\sim d_{P^\star}^\pi}[\|\phi^\star(s,a)\|_{\Sigma_{\rho,\phi^\star}^{-1}}]$. This is upper-bounded by

$$\mathbb{E}_{(s,a)\sim d_{P^\star}^\pi}[\|\phi^\star(s,a)\|_{\Sigma_{\rho,\phi^\star}^{-1}}] \le \sqrt{\mathrm{tr}(\mathbb{E}_{(s,a)\sim d_{P^\star}^\pi}[\phi^\star\phi^{\star\top}]\Sigma_{\rho,\phi^\star}^{-1})}$$

$$\le \sqrt{\bar{C}_{\pi^*,\phi^\star}\,\mathrm{tr}(\mathbb{E}_{(s,a)\sim\rho}[\phi^\star\phi^{\star\top}]\Sigma_{\rho,\phi^\star}^{-1})} \qquad \text{(From Lemma 11)}$$

$$\le \sqrt{\bar{C}_{\pi^*,\phi^\star}\,\mathrm{rank}(\Sigma_\rho)/n}.$$

Here, in the last line, by letting the SVD of $\Sigma_\rho = \mathbb{E}_\rho[\phi\phi^\top]$ be $U\tilde{\Sigma}_\rho U^\top$ where $\tilde{\Sigma}_\rho$ is a $d\times d$ diagonal matrix and $U$ is a $d\times d$ orthogonal matrix , we use

$$\mathrm{tr}\left(\Sigma_\rho\Sigma_{\rho,\phi^\star}^{-1}\right) = \mathrm{tr}(U\tilde{\Sigma}_\rho U^\top\{nU\tilde{\Sigma}_\rho U^\top + \lambda I\}^{-1}) = \mathrm{tr}(\tilde{\Sigma}_\rho U^\top\{nU\tilde{\Sigma}_\rho U^\top + \lambda I\}^{-1}U)$$

$$= \mathrm{tr}(\tilde{\Sigma}_\rho U^\top\{U\{n\tilde{\Sigma}_\rho + \lambda I\}U^\top\}^{-1}U)$$

$$= \mathrm{tr}(\tilde{\Sigma}_\rho U^\top U\{n\tilde{\Sigma}_\rho + \lambda I\}^{-1}U^\top U)$$

$$= \mathrm{tr}(\tilde{\Sigma}_\rho\{n\tilde{\Sigma}_\rho + \lambda I\})^{-1} \le \mathrm{rank}(\Sigma_\rho)/n.$$

Hence, when $P \in \mathcal{M}_\mathcal{D}$, by setting $\lambda$ s.t. $\lambda d \lesssim n\omega_\pi\xi$, we have

$$\mathbb{E}_{(s,a)\sim d_{P^\star}^\pi}[f(s,a)] \le \sqrt{\frac{\gamma\bar{C}_{\pi^*,\phi^\star}\,\mathrm{rank}(\Sigma_\rho)\omega_\pi\ln(|\mathcal{M}|/\delta)}{n}} + \sqrt{\frac{(1-\gamma)\omega_\pi\ln(|\mathcal{M}|/\delta)}{n}}.$$

We use (32) here.

Finally,

$$V_{P^\star}^{\pi^*} - V_{P^\star}^{\hat\pi}$$

$$\le V_{P^\star}^{\pi^*} - \min_{P\in\mathcal{M}_\mathcal{D}} V_P^{\pi^*} \qquad \text{(Recall the proof of the third step in the proof of Theorem 1)}$$

$$\le (1-\gamma)^{-2}\mathbb{E}_{s,a\sim d_{P^\star}^{\pi^*}}\mathrm{TV}(P'(s,a), P^\star(\cdot\mid s,a)) \qquad (P' = \arg\min_{P\in\mathcal{M}_\mathcal{D}} V_P^{\pi^*})$$

$$\lesssim \sqrt{\frac{\bar{C}_{\pi^*,\phi^\star}\,\mathrm{rank}(\Sigma_\rho)\omega_{\pi^*}\ln(|\mathcal{M}|/\delta)}{n(1-\gamma)^4}}.$$

$\square$

The following inequality is an important lemma to connect $\mathbb{E}_{(s,a)\sim d_{P^\star}^\pi}\{f(s,a)\}$ with an elliptical potential $\mathbb{E}_{(\tilde{s},\tilde{a})\sim d_{P^\star}^\pi}\|\phi^\star(\tilde{s},\tilde{a})\|_{\Sigma_{\rho,\phi^\star}^{-1}}$.

**Lemma 4** (One-step back inequality). *Take any $f \subset \mathcal{S}\times\mathcal{A}\to\mathbb{R}$ s.t. $\|f\|_\infty \le B$ and $0 < \lambda \in \mathbb{R}$. Letting $\omega = \max_{s,a}(\pi(a\mid s)/\pi_b(a\mid s))$, for any policy $\pi$, we have*

$$|\mathbb{E}_{(s,a)\sim d_{P^\star}^\pi}\{f(s,a)\}| \le \mathbb{E}_{(\tilde{s},\tilde{a})\sim d_{P^\star}^\pi}\|\phi^\star(\tilde{s},\tilde{a})\|_{\Sigma^{-1}}\sqrt{\{n\omega_\pi\gamma\mathbb{E}_{(s,a)\sim\rho}[f^2(s,a)]\} + \gamma^2\lambda dB^2}$$

$$+ \sqrt{(1-\gamma)\omega_\pi\mathbb{E}_{(s,a)\sim\rho}[f^2(s,a)]}.$$

*where $\Sigma = n\mathbb{E}_{(s,a)\sim\rho}[\phi^\star(s,a)\phi^{\star\top}(s,a)] + \lambda I$.*

*Proof of Lemma 4.* First, we have an equality:

$$\mathbb{E}_{(s,a)\sim d_{P^\star}^\pi}\{f(s,a)\} = \gamma\mathbb{E}_{(\tilde{s},\tilde{a})\sim d_{P^\star}^\pi, s\sim P^\star(\tilde{s},\tilde{a})}\{f(s,a)\} + (1-\gamma)\mathbb{E}_{s\sim d_0, a\sim\pi(s_0)}\{f(s,a)\}.$$

$$(33)$$

The second term in (33) is upper-bounded by

$$\mathbb{E}_{s \sim d_0, a \sim \pi(s_0)} \{f(s,a)\} \leq \mathbb{E}_{s \sim d_0, a \sim \pi(s_0)} \{f^2(s,a)\}\}^{1/2} = \sqrt{\omega_\pi \mathbb{E}_{(s,a) \sim \rho} [f^2(s,a)] / (1-\gamma)}.$$

Next we consider the first term in (33). By CS inequality, we have

$$\left| \mathbb{E}_{(\tilde{s},\tilde{a}) \sim d^\pi_{P^\star}, s \sim P^\star(\tilde{s},\tilde{a})} \{f(s,a)\} \right| = \left| \mathbb{E}_{(\tilde{s},\tilde{a}) \sim d^\pi_{P^\star}} \phi^\star(\tilde{s},\tilde{a})^\top \int \hat{\mu}(s) \pi(a \mid s) f(s,a) d(s,a) \right|$$

$$\leq \mathbb{E}_{(\tilde{s},\tilde{a}) \sim d^\pi_{P^\star}} \|\phi^\star(\tilde{s},\tilde{a})\|_{\Sigma^{-1}_{\rho,\phi^\star}} \| \int \hat{\mu}(s) \pi(a \mid s) f(s,a) d(s,a) \|_{\Sigma_{\rho,\phi^\star}}.$$

Then,

$$\| \int \hat{\mu}(s) \pi(a \mid s) f(s,a) d(s,a) \|^2_{\Sigma_{\rho,\phi^\star}}$$

$$\leq \left\{ \int \hat{\mu}(s) \pi(a \mid s) f(s,a) d(s,a) \right\}^\top \left\{ n \mathbb{E}_{(s,a) \sim \rho} [\phi^\star \phi^{\star\top}] + \lambda I \right\} \left\{ \int \hat{\mu}(s) \pi(a \mid s) f(s,a) d(s,a) \right\}$$

$$\leq n \left\{ \mathbb{E}_{(\tilde{s},\tilde{a}) \sim \rho} \left[ \int \hat{\mu}(s)^\top \phi^\star(\tilde{s},\tilde{a}) \pi(a \mid s) f(s,a) d(s,a) \right] \right\}^2 + B^2 \lambda d$$

$$\text{(Use the assumption } \|f(s,a)\|_\infty \leq B \text{ and } \| \int \hat{\mu}(s) \mathrm{d}(s) \|_2 \leq \sqrt{d})$$

$$= n \left\{ \mathbb{E}_{(\tilde{s},\tilde{a}) \sim \rho, s \sim P^\star(\tilde{s},\tilde{a}), a \sim \pi(s)} [f(s,a)] \right\}^2 + B^2 \lambda d$$

$$\leq n \left\{ \mathbb{E}_{(\tilde{s},\tilde{a}) \sim \rho, s \sim P^\star(\tilde{s},\tilde{a}), a \sim \pi(s)} [f^2(s,a)] \right\} + B^2 \lambda d. \qquad \text{(Jensen)}$$

Finally, the the first term in (33) is upper-bounded by

$$n \left\{ \mathbb{E}_{(\tilde{s},\tilde{a}) \sim \rho, s \sim P^\star(\tilde{s},\tilde{a}), a \sim \pi(s)} [f^2(s,a)] \right\} + \lambda d B^2$$

$$\leq n \omega_\pi \left\{ \mathbb{E}_{(\tilde{s},\tilde{a}) \sim \rho, s \sim P^\star(\tilde{s},\tilde{a}), a \sim \pi_b(s)} [f^2(s,a)] \right\} + \lambda d B^2 \qquad \text{(Importance sampling)}$$

$$\leq n \omega_\pi \left\{ \frac{1}{\gamma} \mathbb{E}_{(s,a) \sim \rho} [f^2(s,a)] \right\} + \lambda d B^2. \qquad \text{(Definition of } \rho)$$

The final statement is immediately concluded.

$\square$

### E.7 PROOFS FOR FACTORED MDPS

*Proof of Theorem 3.* We denote the constrained set as $\mathcal{M}_\mathcal{D}$:

$$\mathcal{M}_\mathcal{D} = \left\{ P = \prod_i P_i \mid \mathbb{E}_\mathcal{D} \left[ \mathrm{TV}(\widehat{P}_{\mathrm{MLE},i}(\cdot \mid s[pa_i], a), P_i(\cdot \mid s[pa_i], a))^2 \right] \leq \xi_i, \forall i \in [1, \cdots, d] \right\}.$$

Following the first step in the proof of Corollary 1, with probability $1 - \delta$, the product $\prod_i P_i^\star$ is in $\mathcal{M}_\mathcal{D}$, i.e.,

$$\mathbb{E}_{s,a \sim \mathcal{D}} \left[ \mathrm{TV}(\widehat{P}_{\mathrm{MLE},i}(\cdot \mid s[pa_i], a), P_i^\star(\cdot \mid s[pa_i], a))^2 \right] \leq \xi_i, \forall i \in [1, \cdots, d], \quad \xi_i = \sqrt{\frac{L_i \log(L_i d/\delta)}{n}}.$$

Note $d$ comes from the union bound. Besides, following the second step in the proof of Corollary 1, for any $P \in \mathcal{M}_\mathcal{D}$, with probability $1 - \delta$,

$$\mathbb{E}_{s,a \sim \rho} \left[ TV(\widehat{P}_i(\cdot \mid s[pa_i], a), P_i^\star(\cdot \mid s[pa_i], a))^2 \right] \leq \xi_i, \forall i \in [1, \cdots, d].$$

After conditioning on the above two events, then, for any $P \in \mathcal{M}_\mathcal{D}$ and $\pi^\star \in \Pi$, we have

$$V_{P^\star}^{\pi^*} - V_P^{\pi^*} \le (1-\gamma)^{-2} \mathbb{E}_{(s,a) \sim d_{P^\star}^{\pi^*}}[\mathrm{TV}(P(\cdot \mid s,a), P^\star(\cdot \mid s,a))]$$

$$\text{(Simulation lemma, Lemma 5)}$$

$$\le (1-\gamma)^{-2} \mathbb{E}_{(s,a) \sim d_{P^\star}^{\pi^*}}\left[\sum_i \mathrm{TV}(P_i(\cdot \mid s[pa_i], a), P_i^\star(\cdot \mid s[pa_i], a))\right]$$

$$\le (1-\gamma)^{-2} \sum_i \sqrt{\mathbb{E}_{(s,a) \sim \rho}\left[\left(\frac{d_{P^\star}^{\pi^*}(s[pa_i], a)}{\rho(s[pa_i], a)}\right)^2\right] \mathbb{E}_{(s,a) \sim \rho}[\mathrm{TV}(P_i(\cdot \mid s[pa_i], a), P_i^\star(\cdot \mid s[pa_i], a))^2]}$$

$$\text{(CS inequality)}$$

$$\le (1-\gamma)^{-2} \sum_i \sqrt{\ddot{C}_{\pi^*,\infty} \mathbb{E}_{(s,a) \sim \rho}[\mathrm{TV}(P_i(\cdot \mid s,a), P_i^\star(\cdot \mid s,a))^2]} \le (1-\gamma)^{-2} \sum_i \sqrt{\ddot{C}_{\pi^*,\infty} \xi_i}$$

$$\le (1-\gamma)^{-2} \sqrt{d \ddot{C}_{\pi^*,\infty} \sum_i \xi_i}$$

$$\text{(CS inequality)}$$

$$\le c(1-\gamma)^{-2} \sqrt{d \ddot{C}_{\pi^*,\infty} \frac{L \ln(Lnd/\delta)}{n}}.$$

Here, recall

$$\ddot{C}_{\pi^*,\infty} = \max_{i \in [1,\cdots,d]} \mathbb{E}_{(s,a) \sim \rho}\left[\left(\frac{d_{P^\star}^{\pi^*}(s[pa_i], a)}{\rho(s[pa_i], a)}\right)^2\right].$$

Following the third step in the proof of Corollary 1, the statement is concluded. □

Next, we show that $\ddot{C}_{\pi^*,\infty} \le C_{\pi^*,P^\star} = \max_{s,a} \frac{d_{P^\star}^{\pi^*}(s,a)}{\rho(s,a)}$.

**Proposition 1** (Comparison of $L^\infty$-density-ratio based concentrabiliity coefficient between factored MDPs and non-factored MDPs ). *For any $\pi^*$, we have:*

$$\ddot{C}_{\pi^*,\infty} \le C_{\pi^*,\infty}. \tag{34}$$

*Proof.* From now on, for any $i \in [1,\cdots,d]$, by defining $\mathcal{S}_i'$ s.t. $\mathcal{S} = \mathcal{S}_i \times \mathcal{S}_i'$, we prove

$$\max_{s_i \in \mathcal{S}_i, a \in \mathcal{A}} \frac{d_{P^\star}^{\pi^*}(s_i, a)}{\rho(s_i, a)} \le \max_{s \in \mathcal{S}_i, s_i' \in \mathcal{S}_i, a \in \mathcal{A}} \frac{d_{P^\star}^{\pi^*}(s_i, s_i', a)}{\rho(s_i, s_i', a)} = C_{\pi^*,\infty}.$$

Then, (34) is easily proved.

First, for any $s_i \in \mathcal{S}_i, a \in \mathcal{A}$, we have

$$\max_{s_i'} \frac{d_{P^\star}^{\pi^*}(s_i, s_i', a)}{\rho(s_i, s_i', a)} = \max_{s_i'} \frac{d_{P^\star}^{\pi^*}(s_i, a) d_{P^\star}^{\pi^*}(s_i' \mid s_i, a)}{\rho(s_i, a) \rho(s_i' \mid s_i, a)} = \frac{d_{P^\star}^{\pi^*}(s_i, a)}{\rho(s_i, a)} \max_{s_i'} \frac{d_{P^\star}^{\pi^*}(s_i' \mid s_i, a)}{\rho(s_i' \mid s_i, a)} \ge \frac{d_{P^\star}^{\pi^*}(s_i, a)}{\rho(s_i, a)}. \tag{35}$$

Here, we use

$$1 \le \max_{s_i'} \frac{d_{P^\star}^{\pi^*}(s_i' \mid s_i, a)}{\rho(s_i' \mid s_i, a)},$$

which is proved by the contradiction argument, that is, if $1 > \max_{s_i'} \frac{d_{P^\star}^{\pi^*}(s_i' \mid s_i, a)}{\rho(s_i' \mid s_i, a)}$, both $\rho$ and $d_{P^\star}^{\pi^*}$ cannot be probability mass functions since we would get

$$1 = \sum_{s_i'} d_{P^\star}^{\pi^*}(s_i' \mid s_i, a) \le \max_{s_i'} \left(\frac{d_{P^\star}^{\pi^*}(s_i' \mid s_i, a)}{\rho(s_i' \mid s_i, a)}\right) \sum_{s_i'} \rho(s_i' \mid s_i, a) < \sum_{s_i'} \rho(s_i' \mid s_i, a).$$

Then, by taking the maximum over $s_i \in \mathcal{S}_i, a \in \mathcal{A}$ for both sides on (35), we have

$$\max_{s_i, a} \frac{d_{P^\star}^{\pi^*}(s_i, a)}{\rho(s_i, a)} \le \max_{s_i, s_i', a} \frac{d_{P^\star}^{\pi^*}(s_i, s_i', a)}{\rho(s_1, s_i', a)}.$$

□

## F  AUXILIARY LEMMAS

**Lemma 5** (Simulation Lemma). *Consider any two transitions $P$ and $\widehat{P}$, and any policy $\pi : \mathcal{S} \to \Delta(\mathcal{A})$. We have:*

$$|V_P^\pi - V_{\widehat{P}}^\pi| \le |(1-\gamma)^{-1} \mathbb{E}_{s,a \sim d_P^\pi}[\mathbb{E}_{s' \sim P(s,a)}[V_{\widehat{P}}^\pi(s')] - \mathbb{E}_{s' \sim P(s,a)}[V_{\widehat{P}}^\pi(s')]]|$$

$$\le (1-\gamma)^{-2} \mathbb{E}_{s,a \sim d_P^\pi}\left[\text{TV}(P(\cdot|s,a), \widehat{P}(\cdot|s,a))\right].$$

*Proof.* Such simulation lemma is standard in model-based RL literature and the derivation can be found, for instance, in the proof of Lemma 10 from Sun et al. (2019). □

**Lemma 6** (MLE guarantee). *Given a set of models $\mathcal{M} = \{P : \mathcal{S} \times \mathcal{A} \to \Delta(\mathcal{S})\}$ with $P^\star \in \mathcal{M}$, and a dataset $\mathcal{D} = \{s_i, a_i, s_i'\}_{i=1}^n$ with $s_i, a_i \sim \rho$, and $s_i' \sim P^\star(s_i, a_i)$, let $\widehat{P}_{\text{MLE}}$ be*

$$\widehat{P}_{\text{MLE}} = \underset{P \in \mathcal{M}}{\arg\min} \sum_{i=1}^n - \ln P(s_i'|s_i, a_i).$$

*With probability at least $1 - \delta$, we have:*

$$\mathbb{E}_{s,a \sim \rho} \text{TV}(\widehat{P}_{\text{MLE}}(\cdot|s,a), P^\star(\cdot|s,a))^2 \lesssim \frac{\ln(|\mathcal{M}|/\delta)}{n}.$$

*Proof.* Refer to (Agarwal et al., 2020b, Section E) □

**Lemma 7** (MLE guarantee for tabular models).

$$\mathbb{E}_{\mathcal{D}}\left[\text{TV}(P(\cdot|s,a), \widehat{P}_{\text{MLE}}(\cdot|s,a))^2\right] \le \frac{|\mathcal{S}||\mathcal{A}|\{|\mathcal{S}| \ln 2 + \ln(2|\mathcal{S}||\mathcal{A}|/\delta)\}}{2n}.$$

*Proof.* From Chang et al. (2021, Lemma 12) , with probability $1 - \delta$,

$$\text{TV}(P(\cdot|s,a), \widehat{P}_{\text{MLE}}(\cdot|s,a))^2 \le \frac{|\mathcal{S}| \ln 2 + \ln(2|\mathcal{S}||\mathcal{A}|/\delta)}{2N(s,a)} \quad \forall (s,a) \in \mathcal{S} \times \mathcal{A},$$

where $N(s,a)$ is the number of visiting times for $(s,a)$. Then,

$$\mathbb{E}_{\mathcal{D}}\left[\text{TV}(P(\cdot|s,a), \widehat{P}_{\text{MLE}}(\cdot|s,a))^2\right] \le \mathbb{E}_{\mathcal{D}}\left[\frac{|\mathcal{S}| \ln 2 + \ln(2|\mathcal{S}||\mathcal{A}|/\delta)}{2N(s,a)}\right]$$

$$\le \sum_{(s,a)}\left[\frac{|\mathcal{S}| \ln 2 + \ln(2|\mathcal{S}||\mathcal{A}|/\delta)}{2n}\right]$$

$$= \frac{|\mathcal{S}||\mathcal{A}|\{|\mathcal{S}| \ln 2 + \ln(2|\mathcal{S}||\mathcal{A}|/\delta)\}}{2n}.$$

□

**Lemma 8** (MLE guarantee for KNRs).

$$\left\|\left(\widehat{W}_{\text{MLE}} - W^\star\right)(\Sigma_n)^{1/2}\right\|_2 \le \beta_n.$$

*Proof.* The proof directly follows the confidence ball construction and proof from (Kakade et al., 2020). □

**Lemma 9** ($\ell_1$ Distance between two Gaussians). *Consider two Gaussian distributions $P_1 := \mathcal{N}(\mu_1, \zeta^2 \mathbf{I})$ and $P_2 := \mathcal{N}(\mu_2, \zeta^2 \mathbf{I})$. We have:*

$$\text{TV}(P_1, P_2) \le \frac{1}{\zeta} \|\mu_1 - \mu_2\|_2.$$

*Proof.* This lemma is proved by Pinsker's inequality and the closed-form of the KL divergence between $P_1$ and $P_2$. Refer to (Kakade et al., 2020). □

**Lemma 10** (Property of linear mixture MDPs). *Let $P(\theta) = \theta^\top \psi(s, a, s')$. Suppose $P(\theta) \in \mathcal{S} \times \mathcal{A} \to \Delta(\mathcal{S})$. For any function $V \in \mathcal{S} \to [0, 1]$, letting $\psi_V(s, a) = \int \psi(s, a, s') V(s') \mathrm{d}(s')$, we suppose $\|\psi_V(s, a)\|_2 \leq 1$. The following theorems hold:*

1. *For any $(s, a, s')$, we have $|P(\theta)(s, a, s') - P(\theta')(s, a, s')| \leq \|\theta - \theta'\|_2$.*

2. *For any $(s, a)$, we have $\mathrm{TV}(P(\theta)(s, a, \cdot), P(\theta')(s, a, \cdot)) \leq \|\theta - \theta'\|_2$. Besides, for any $V : \mathcal{S} \to [0, 1]$, we have*

$$|(\theta - \theta')\psi_V(s, a)| \leq \mathrm{TV}(P(\theta)(s, a, \cdot), P(\theta')(s, a, \cdot)).$$

3.

$$C_{\pi^\star, P^\star}^\dagger = \sup_x \frac{x^\top \mathbb{E}_{(s,a) \sim d_{P^\star}^{\pi^*}}[\psi_{V_{(s,a,x)}}(s, a) \psi_{V_{(s,a,x)}}^\top(s, a)] x}{x^\top \mathbb{E}_{(s,a) \sim \rho}[\psi_{V_{(s,a,x)}}(s, a) \psi_{V_{(s,a,x)}}^\top(s, a)] x},$$

$$V_{(s,a,x)} = \arg\max_{V:\mathcal{S} \to [0,1]} \left| x^\top \int \phi(s, a, s') V(s') \mathrm{d}(s') \right|.$$

4. *In linear MDPs (i.e., $\psi(s, a, s') = \phi(s, a) \otimes \mu(s')$), we have*

$$\sup_{V \in \{\mathcal{S} \to [0,1]\}} \sup_x \frac{x^\top \mathbb{E}_{(s,a) \sim d_{P^\star}^{\pi^*}}[\psi_V(s, a) \psi_V^\top(s, a)] x}{x^\top \mathbb{E}_{(s,a) \sim \rho}[\psi_V(s, a) \psi_V^\top(s, a)] x} = \sup_x \frac{x^\top \mathbb{E}_{d_{P^\star}^{\pi^*}}[\phi(s, a) \phi(s, a)^\top] x}{x^\top \mathbb{E}_\rho[\phi(s, a) \phi(s, a)^\top] x}.$$

*Proof.* We prove the first statement. This is proved by

$$|P(\theta) - P(\theta')| = |(\theta - \theta')\psi(s, a, s')| \leq \|\theta - \theta'\|_2 \|\psi(s, a, s')\|_2 \leq \|\theta - \theta'\|_2,$$

Here, we use $\|\psi(s, a, s')\|_2 \leq 1$ which is proved by the assumption by setting $V(s) = I(s' = s)$ for any $s'$.

Next, we prove the second statement. For fixed $\theta \in \mathbb{R}^d$ and $(s, a) \in \mathcal{S} \times \mathcal{A}$, we have

$$\mathrm{TV}(P(\theta)(s, a, \cdot), P(\theta^\star)(s, a, \cdot)) = \sup_{V:\mathcal{S} \to [0,1]} \left| \int (\theta - \theta^\star)^\top \psi(s, a, s') V(s') \mathrm{d}(s') \right|$$

$$= \sup_{V:\mathcal{S} \to [0,1]} \left| (\theta - \theta^\star)^\top \int \psi(s, a, s') V(s') \mathrm{d}(s') \right|$$

$$= \left| (\theta - \theta^\star)^\top \int \psi(s, a, s') V_{(s,a,\theta)}(s') \mathrm{d}(s') \right|$$

$$= \left| (\theta - \theta^\star)^\top \psi_{V_{(s,a,\theta)}}(s, a) \right|.$$

In the third line, we define $V(s, a, \theta) = \arg\max_{V:\mathcal{S} \to [0,1]} |(\theta - \theta^\star)^\top \int \psi(s, a, s') V(s') \mathrm{d}(s')|$.

Then, from CS inequality,

$$\mathrm{TV}(P(\theta)(s, a, \cdot), P(\theta^\star)(s, a, \cdot)) \leq \|(\theta - \theta^\star)\|_2 \|\psi_{V_{(s,a,\theta)}}(s, a)\|_2 \leq \|\theta - \theta^\star\|_2.$$

We use the assumption $\|\psi_{V_{(s,a,\theta)}}(s, a)\|_2 \leq 1$. This concludes the second statement. Besides, for any $V : \mathcal{S} \to [0, 1]$, we have

$$|(\theta - \theta')\psi_V(s, a)| \leq |(\theta - \theta^\star)^\top \psi_{V_{(s,a,\theta)}}(s, a)|$$
$$\leq \mathrm{TV}(P(\theta)(s, a, \cdot), P(\theta')(s, a, \cdot)).$$

The third statement is immediately concluded by

$$\frac{\mathbb{E}_{(s,a) \sim d_{P^\star}^{\pi^*}}[\mathrm{TV}(P(\theta)(s, a, \cdot), P(\theta^\star)(s, a, \cdot))^2]}{\mathbb{E}_{(s,a) \sim \rho}[\mathrm{TV}(P(\theta)(s, a, \cdot), P(\theta^\star)(s, a, \cdot))^2]} = \frac{\mathbb{E}_{(s,a) \sim d_{P^\star}^{\pi^*}}[|(\theta - \theta^\star)^\top \psi_{V_{(s,a,\theta)}}(s, a)|^2]}{\mathbb{E}_{(s,a) \sim \rho}[|(\theta - \theta^\star)^\top \psi_{V_{(s,a,\theta)}}(s, a)|^2]}. \quad (36)$$

Finally, we prove the fourth statement. Suppose $\psi(s, a, s') = \phi(s, a) \otimes \mu(s')$ ($\otimes$ denotes kronerker product). Then, $\phi_V(s, a, s') = \phi(s, a) \otimes \int \mu(s') V(s') \mathrm{d}(s')$. Then, by defining a vector $\mu(V) = \int \mu(s') V(s') \mathrm{d}(s')$, we immediately have

$$\frac{x^\top \mathbb{E}_{(s,a) \sim d_{P^\star}^{\pi^*}}[\psi_V(s, a) \psi_V^\top(s, a)] x}{x^\top \mathbb{E}_{(s,a) \sim \rho}[\psi_V(s, a) \psi_V^\top(s, a)] x} = \sup_x \frac{x^\top \mathbb{E}_{(s,a) \sim d_{P^\star}^{\pi^*}}[(\phi(s, a) \otimes \mu(V))(\phi(s, a) \otimes \mu(V))^\top] x}{x^\top \mathbb{E}_{(s,a) \sim \rho}[(\phi(s, a) \otimes \mu(V))(\phi(s, a) \otimes \mu(V))^\top] x}.$$
(37)

Here, we have

$$\mathbb{E}_\rho[(\phi(s, a) \otimes \mu(V))(\phi(s, a) \otimes \mu(V))^\top] = \mathbb{E}_\rho[(\phi(s, a) \otimes \mu(V))(\phi(s, a)^\top \otimes \mu(V)^\top)]$$
$$= \mathbb{E}_\rho[(\phi(s, a) \phi(s, a)^\top)] \otimes (\mu(V) \mu(V)^\top).$$

We notice

$$\{\mathbb{E}_\rho[(\phi(s, a) \phi(s, a)^\top)] \otimes (\mu(V) \mu(V)^\top)\}^{1/2} = \mathbb{E}_\rho[\phi(s, a) \phi(s, a)^\top]^{1/2} \otimes (\mu(V) \mu(V)^\top)^{1/2}.$$

This is because the square root of a matrix is unique and we have $(A^{1/2} \otimes B^{1/2})(A^{1/2} \otimes B^{1/2}) = AB$ for symmetric matrices $A$ and $B$. Then, by denoting $F_\rho = \mathbb{E}_\rho[\phi(s, a) \phi(s, a)^\top]$, $F_{d_{P^\star}^\pi} = \mathbb{E}_{d_{P^\star}^\pi}[\phi(s, a) \phi(s, a)^\top]$ and denoting the pseudo inverse of $F$ as $F^+$, we can see (37) is equal to

$$\{F_\rho^{1/2} \otimes (\mu(V) \mu(V)^\top)^{1/2}\}^+ \{F_{d_{P^\star}^\pi} \otimes (\mu(V) \mu(V)^\top)\} \{F_\rho^{1/2} \otimes (\mu(V) \mu(V)^\top)^{1/2}\}^+$$
$$= \{F_\rho^{-1/2} \otimes (\mu(V) \mu(V)^\top)^{-1/2}\} \{F_{d_{P^\star}^\pi} \otimes (\mu(V) \mu(V)^\top)\} \{F_\rho^{-1/2} \otimes (\mu(V) \mu(V)^\top)^{-1/2}\}$$
$$= \{F_\rho^{-1/2} F_{d_{P^\star}^\pi} F_\rho^{-1/2}\} \otimes \{(\mu(V) \mu(V)^\top)^{-1/2} (\mu(V) \mu(V)^\top)(\mu(V) \mu(V)^\top)^{-1/2}\}$$
$$= \{F_\rho^{-1/2} F_{d_{P^\star}^\pi} F_\rho^{-1/2}\} \otimes I_k \ (k = \mathrm{rank}(\mu(V) \mu(V)^\top)).$$

Here, $I_k$ is a diagonal matrix s.t. $k \in \mathbb{N}^+$ values in the diagonal entries are 1 and the rest of values are 0. Then, the maximum singular value of $\{F_\rho^{-1/2} F_{d_{P^\star}^\pi} F_\rho^{-1/2}\} \otimes I_k$ is equal to the one of $\{F_\rho^{-1/2} F_{d_{P^\star}^\pi} F_\rho^{-1/2}\}$. This is equal to

$$\sup_x \frac{x^\top F_{d_{P^\star}^\pi} x}{x^\top F_\rho x}$$

Hence, the fourth statement is concluded.

$\square$

**Lemma 11** (Distribution shift lemma). *Suppose $A_1, A_2, A_3$ are semipositive definite matrices:*

$$\mathrm{Tr}(A_1 A_2) \leq \sigma_{\max}(A_3^{-1/2} A_1 A_3^{-1/2}) \, \mathrm{Tr}(A_3 A_2).$$

*Note*

$$\sigma_{\max}(A_3^{-1/2} A_1 A_3^{-1/2}) = \sup_{x \in \mathbb{R}^d} \frac{x^\top A_1 x}{x^\top A_3 x}.$$

*Proof.*

$$\mathrm{Tr}(A_1 A_2) = \mathrm{Tr}(A_1^{1/2} A_2 A_1^{1/2}) = \mathrm{Tr}(A_1^{1/2} A_3^{-1/2} A_3^{1/2} A_2 A_3^{1/2} A_3^{-1/2} A_1^{1/2})$$
$$= \mathrm{Tr}(A_3^{-1/2} A_1 A_3^{-1/2} A_3^{1/2} A_2 A_3^{1/2}).$$

In addition, for any semipositive definite matrices $A, B$ we have

$$\mathrm{Tr}(AB) = \mathrm{Tr}(U \Lambda U^\top B) = \mathrm{Tr}(\Lambda U^\top B U) \leq \sigma_{\max}(\Lambda) \, \mathrm{Tr}(U^\top B U) = \sigma_{\max}(A) \, \mathrm{Tr}(B),$$

where $U \Lambda U^\top$ is the SVD decomoposition of $A$. This concludes that

$$\mathrm{Tr}(A_1 A_2) \leq \sigma_{\max}(A_3^{-1/2} A_1 A_3^{-1/2}) \, \mathrm{Tr}(A_3 A_2).$$

$\square$

The following lemma is useful to obtain the generalized result of Theorem 1. The proof is given in (Wainwright, 2019, Theorem 3.27). We first define

$$Z = \sup_{f \in \mathcal{F}} |\{\mathbb{E}_{\mathcal{D}} - \mathbb{E}_\rho\}[f]$$

$$\Sigma^2 = \sup_{f \in \mathcal{F}} \mathbb{E}_{\mathcal{D}}[\{f(s,a) - \mathbb{E}_\rho[f(s,a)]\}^2], \ \sigma^2 = \sup_{f \in \mathcal{F}} \text{var}[f(s,a)].$$

**Lemma 12** (Functional Bernstein's inequality: Talagrand concentration inequality for empirical process). *Suppose $\|f\|_\infty \leq B$. With probability $1 - \delta$,*

$$|Z - \mathbb{E}[Z]| \leq \Sigma^2 \sqrt{\frac{\log(c/\delta)}{n}} + \frac{B \log(c/\delta)}{n}.$$

*As an immediate corollary,*

$$|Z - \mathbb{E}[Z]| \leq \{\sigma^2 + B\mathbb{E}[Z]\} \sqrt{\frac{\log(c/\delta)}{n}} + \frac{B \log(c/\delta)}{n}.$$

