# OpenReview forum: "Pessimistic Model-based Offline Reinforcement Learning under Partial Coverage"
_ICLR.cc/2022/Conference — ICLR 2022 Poster_

### Official Review · Reviewer_Tgsy · 2021-10-29

**Correctness:** 4
**Technical Novelty And Significance:** 2
**Empirical Novelty And Significance:** Not applicable
**Recommendation:** 6
**Confidence:** 3

**Main Review:**

Offline RL is a hot topic nowadays and relaxing the full coverage condition in theory is important. Overall, I feel this paper contains lots valuable results and is the first of such kind of guarantee for model-based algorithm+partial coverage. But more or less I feel a bit less exciting after reading the paper since it's hard to get the strong point. It should not be the case that If there exist guarantees for model-free algorithm under scenario A and B, we must have guarantees for model-based algorithm under scenarios A and B. I would like to gain some insights on why we want to study model-based algorithms in this offline setting over a model-free algorithm. Pessimism has been studied in quite a few other papers and so is the partial coverage condition.

Some specific comments:

1. One concern is about the algorithm. I do not feel it's a real algorithm and can be implemented in an efficient way. I hope the practitioner can gain some insights on algorithm design after reading the theory paper. In particular, even for linear mixture MDPs, you need to integrate over the whole state space which loses the computational efficiency. Please correct me if I am wrong. And no experiments which are a bit disappointed. It will be good to see the comparison between model-based algorithm and model-free algorithm with pessimism.

2. I appreciate the authors would like to include as many examples as possible but some of them are not quite well-explained and the paper now is a bit dense. For example, the offline representation learning with a low-rank structure is not well-motivated. And there is no even motivation why people should study this. Is there any technical challenge? If so, it should be discussed clearly. Again, it might be hard to get the key point since for each setting, the rate has been studied and is not supervised.

3. It may not be fair to say model-free offline RL approaches often require additional structural conditions in the function class (e.g., Bellman completion). The linear mixture MDPs might be a worse condition since it compresses too much information and there is no empirical evidence to justify it as far as I know. It's purely invented to prove something.

4. About the function class. What do you mean by "general function class"? What's your function class complexity measure, like VC-dimension or Rademacher complexity or else? In Theorem 1, I just saw |M|. Does it mean M has to be finite?

**Summary Of The Paper:**

This paper studied offline RL for policy optimization and proposed a model-based algorithm with guarantees that can hold under partial coverage. The results can cover several existing popular model classes.

**Summary Of The Review:**

This paper provides rich theoritical results for the important offline RL problem but it's a bit lacking a strong exciting point.

---

> ### Author Response · Authors · 2021-11-12
> **Rebuttal for Tgsy**
>
> Thank you for your detailed comments!
>
> Q: It should not be the case that If there exist guarantees for model-free algorithm under scenario A and B, we must have guarantees for model-based algorithm under scenarios A and B. I would like to gain some insights on why we want to study model-based algorithms in this offline setting over a model-free algorithm.
>
> We can confidently assure our paper is not one of the papers moving the existing idea of model-free to model-based. We would like to clarify existing mode-free papers can mainly handle linear models. Incorporating any function approximation beyond linear models like our paper (with PAC guarantee) is a highly challenging task. We chose the model-based setting because there is a growing number of many offline RL works with general function approximation, but all of them fail to show the PAC guarantee under partial coverage.
>
> Q: One concern is about the algorithm. I do not feel it can be implemented in an efficient way.. In particular, even for linear mixture MDPs...
>
> A:  This is an important suggestion. We will add more discussion. While we agree the current algorithm is not computationally efficient in general, this is because we are trying to tackle general MDPs requiring general function approximation while prior arts (with PAC guarantee) focused on tabular and linear function approximation (Jin et al. (2020b)). Devising a computationally efficient algorithm that works for any MDPs under partial coverage with general function approximation is important but challenging as one has to deal with nonlinear and nonconvex optimization.
>
> We potentially could make CPPO computationally tractable by using an actor-critic formulation.  First, we can use natural policy gradient (softmax policy update) for the policy optimization part.  Second, while the current optimization form in CPPO is constraint optimization, it has been also formulated as the minimization with the penalty term (adding a Lagrange multiplier) . These strategies are employed in model-free algorithms such as Zanette et al. (2021); Xie et al. (2021).
>
>
> Finally, note though linear mixture MDPs might be not practical, our algorithm can accommodate any models including more practical models such as KNRs/GPs and low-rank MDPs.
>
>
> Q:  Because of many examples, the paper now is a bit dense. E.g. the offline representation learning with a low-rank structure is not well-motivated. Is there any technical challenge? The rate has been studied and is not supervised.
>
> A:  Thank you for the good suggestion. We would try to include more motivation for each model.  Due to the page constraint, we might not be fully able to include them. Low-rank MDPs ( note they are different from linear MDPs) include latent variable models, which are used in a number of papers [Mistra et.al 2020, Kaiser et. al (2020)].  Low-rank MDPs are technically challenging since we cannot use linear function approximation anymore (features are unknown a priori). We will give more motivation in the revision.
>
> We will also try to include more implications of the rate. Since our main focus is about partial coverage, we focus on the implication associated with partial coverage. But, we agree that the rate is also important.
>
> Misra et.al (2020) Kinematic state abstraction and provably efficient rich-observation reinforcement learning.
>
> Kaiser et. al (2020). Model-based reinforcement learning for atari.
>
> Q: It may not be fair to say model-free offline RL approaches often require additional structural conditions in the function class (e.g., Bellman completion). The linear mixture MDPs might be a worse condition and there is no empirical evidence to justify it.
>
> A: We can understand your concern about linear mixture MDPs being artificial. But this discussion is tangential to our paper since we just use linear mixture MDPs as an example to show our general theory works for existing models. More importantly, our CPPO works on many practically used models such as LQRs, KNRs, Gaussian process models and Low-rank MDPs.
>
> Q: About the function class. What do you mean by "general function class"?. Does it mean M has to be finite?
>
> A: M does not need to be finite.  On page 5 in the current draft, we mentioned the following.
>
> “Theorem 1 considers the case where the hypothesis class M is finite. When the hypothesis class is infinite, we can still obtain the PAC guarantee by utilizing the generalized result in Section A for any realizable model class with valid statistical complexity (e.g., localized Rademacher complexity).”
>
> In summary, we can use Radecmer complexity. If the VC-dimension is finite,  the Rademacher complexity is upper-bounded by VC dimension as well. The details are in Section A in Appendix.
>
> Finally, note that our complexity depends on $\log(M)$ and this is a standard statistical complexity measure, and such discrete hypothesis class with $\log(M)$ dependence is widely used in bandit and RL literature (Chen &Jiang, 2019).

---

### Official Review · Reviewer_Pn6R · 2021-10-30

**Correctness:** 4
**Technical Novelty And Significance:** 3
**Empirical Novelty And Significance:** 3
**Recommendation:** 6
**Confidence:** 4

**Main Review:**

Strength:
This work contributes to the literature of pessimism principle for offline RL by taking a model-based approach. It is an important complement to model-free type works by showing another way to get things done.
The various examples unveil some untackled problems that can actually be dealt with the model-free framework elegantly.
Comparison of state-of-the-art works is detailed and essential, which might be of use for upcoming works in the field.
The paper is in general clearly written and main points are neatly presented.

Weakness:
The advantage of tackling infinite policy class is a little bit over-advertised. In model-based approach, the hardness of learning with complex policy class naturally takes over to complex model class. In fact, model-free approaches (e.g., Xie et al., 2021) can also deal with infinite policy class with bounded complexity. That’s why I feel the point a bit over-advertised.
Implementation feasibility: Is the model-based approach easily implementable like function approximation with bonus functions (e.g., Jin et al, 2020)? It would be nice to have some discussions on this point.

Other questions:
Dependence on model class complexity: When the model class is infinite, the learning performance of MLE requires complexity measures like bracket numbers, hidden in the \xi_n in Appendix A. Is there any intuitions on how it compares to model-free approach?

**Summary Of The Paper:**

This paper studies the model-based approach to offline RL employing the pessimism principle, which complements the line of research on model-free pessimism. Based on TV distance between the learned MLE model and the realizable ground truth, PAC guarantee of the learned policy is established for various examples, some of which have not been tackled with model-free pessimistic methods.

**Summary Of The Review:**

In general, the paper is well written and main points are neatly addressed. With a model-based approach using concentration properties of MLE, this work contributes to the literature of pessimistic offline RL and shows various concrete examples the framework can deal with. Some weakness include the over-advertisement of tackling infinite policy class, and lack of discussion on feasibility of implementation.

---

> ### Author Response · Authors · 2021-11-12
> **Rebuttal for Pn6R**
>
> We appreciate your feedback!
>
> Q: Weakness: The advantage of tackling infinite policy class is a little bit over-advertised. In model-based approach, the hardness of learning with complex policy class naturally takes over to complex model class. In fact, model-free approaches (e.g., Xie et al., 2021) can also deal with infinite policy class with bounded complexity. That’s why I feel the point a bit over-advertised.
>
> A: We first would like to clarify that our main point here is that CPPO can compete with the policy class containing all stochastic policies, i.e.,  a class with **unbounded statistical complexity**.  Xie et.al 2021 can deal with infinite hypothesis class but only with **bounded statistical complexity**. In appendix E, we included the details where we show that their results depend on $\log |\Pi|$, meaning that $|\Pi|$ needs to have some bounded statistical complexity. Compared to their result, our sample bound is agnostic to  $\log |\Pi|$. So, CPPO can allow for any policy class with possibly infinite bounded complexity such as the class containing all stochastic policies and even non-Markovian policies. Though we might give the impression of stressing this point too much in our draft, we emphasize that it is indeed a crucial point, which can be easily overlooked.
>
> Q: Implementation feasibility: Is the model-based approach easily implementable like function approximation with bonus functions (e.g., Jin et al, 2020)? It would be nice to have some discussions on this point.
>
> A: Thank you for the important suggestion. While we agree the current algorithm is not computationally efficient in general, this is because we are trying to tackle general MDPs requiring general function approximation while prior arts focused on tabular and linear function approximation (Jin et al. (2020b)  Zanette et al. (2021)) . Devising a computationally efficient algorithm that works for any MDPs with general function approximation (with the same statistical guarantee as CPPO) is challenging as one has to deal with nonlinear and nonconvex optimization. We will definitely add more discussion.
>
> We potentially could make CPPO computationally tractable by using an actor-critic formulation.  First, we can use natural policy gradient (softmax policy update) for the policy optimization part.  Second, while the current optimization form in CPPO is constraint optimization, it has been also formulated as the minimization with the penalty term (adding a Lagrange multiplier) . These strategies are employed in model-free algorithms such as Zanette et al. (2021); Xie et al. (2021).
>
> Q: Other questions: Dependence on model class complexity: When the model class is infinite, the learning performance of MLE requires complexity measures like bracket numbers, hidden in the \xi_n in Appendix A. Is there any intuition on how it compares to model-free approach?
>
> This is an interesting point. In the model-free approach, (e.g., Xie et.al 2021 as an example) , we can similarly use the covering number/bracketing number to deal with the infinite hypothesis class. Intuitively, the model-free approach employs the statistical result for regression.  The model-based approach employs the statistical result for MLE. The statistical complexities are similar in the sense that we can use similar devices such as covering number/bracketing numbers; however not exactly comparable either.

---

### Official Review · Reviewer_gN7e · 2021-11-02

**Correctness:** 4
**Technical Novelty And Significance:** 4
**Empirical Novelty And Significance:** Not applicable
**Recommendation:** 8
**Confidence:** 4

**Main Review:**

The work takes a model based perspective, where a general, intractable algorithm is proposed first.

The advantage of model based algorithm in certain applications (for example the lack requirement of Bellman completeness) is well motivated, e.g. remark 2 and also the conclusion.

The bound is then specialized to several well studied setting of interests. Among these, the low-rank setting with unknown features is of particular interests. Per my understanding the computational tractability of the method is not discussed, and the algorithm is therefore not efficient in that respect.

Since the algorithm is model free but many proposals in this space are model-based, this work is a first serious attempt at a offline model based RL with pessimism.

**Summary Of The Paper:**

The paper builds on a recent trend of pessimism for offline RL to return high performance policies on an event with large probability.
The contribution of the paper is theoretical, and about model-based methods.

**Summary Of The Review:**

The paper provides a first good attempt at offline rl with pessimism, and should be accepted.

---

> ### Author Response · Authors · 2021-11-11
> **Rebuttal for gN7e**
>
> Thank you for your positive comments!

---

### Official Review · Reviewer_3dYA · 2021-11-02

**Correctness:** 3
**Technical Novelty And Significance:** 2
**Empirical Novelty And Significance:** Not applicable
**Recommendation:** 5
**Confidence:** 5

**Main Review:**

"worse performance": the theoretical performance provided here is somewhat worse than prior arts. For tabular MDP, many results show a linear dependence of S, which is much better than Corollary 1. In addition, the linear mixture MDP doesn't include linear MDP as a special case, which is misleading. Definition 3 is different from the standard definition of linear MDP, which doesn't assume $\mu$ is known. This is substantial since the dimension of standard linear MDP is $Sd \gg d^2$. Summing up, a linear rate w.r.t. model dimension is achieved for CPPO, which is worse than the general sub-linear rate.

"computational issue": the authors should discuss the implementability of the proposed algorithm, which is missing here. As for me, CPPO is much more impractical than prior arts.

"model based vs. model free": the definitions of model based/model free are strange for me. Many "model free" results stated in this paper are commonly considered as model based methods, such as methods for tabular MDP and linear MDP and so on. Specifically, in tabular MDP, one constructs an empirical MDP with penalized rewards first, and then solves this model to get an optimal policy. In linear MDP, one uses least square to deal with the empirical MDP with penalty. The authors should make clear of this point.

"technique correctness": in Corollary 2, $c_3$ is at most $1/S$, which seems to be considered as a constant here.

**Summary Of The Paper:**

This paper propose an algorithm named Constrained Pessimistic Policy Optimization (CPPO) for offline reinforcement learning, and PAC guarantees are provided for many specialized Markov Decision Processes under the partial coverage assumption of offline data.

**Summary Of The Review:**

Given the above technique issues, I think this paper is not ready for publication.

---

> ### Author Response · Authors · 2021-11-11
> **Rebuttal for 3dYA**
>
> Thank you for your detailed review!
>
> Q: "worse performance": the theoretical performance provided here is worse than prior arts For tabular MDP, many results are much better than Corollary 1.
>
> A:  We appreciate that you bring this potentially confusing important point; however, we humbly disagree with your statement. As far as we know, our result is the state-of-the-art result, even in the tabular case.
>
> First, it is important to recall the output of CPPO can compete with any stochastic policies. These guarantees are obtained in papers like [Zannete et.al (2021), Xie et.al (2021)]; however, not for [Rashidinejad 2021.Jin et.al 2020b]. In the latter papers, the output can only compete with the optimal policy; but not all stochastic policies. This difference is critical since when the optimal policy is not covered by the offline data, the former papers can still state that the output can compete with any policies covered by the offline data; on the other hand, the latter papers cannot ensure the quality of the output policy anymore.
>
>
> Going back to the comparison in terms of the sample complexity, Zannete et.al (2021) and Xie et.al (2021) have the same dependence regarding $|S|$ as our paper.  Thus, our theoretical performance is not worse than the one in their works.  (They derive the rate on linear MDPs (Jin et al. (2020a), and the final order is O(d). Note that their papers hide $\sqrt{d}$ in the empirical covariance matrix term. By using the conversion $d=|S||A|$, we can see their rate is O(|S||A|).
>
> I guess you have the rate $\sqrt{|S|}$ of [Rashidinejad 2021] in your mind. But as mentioned, this paper has a **weaker guarantee**; thus, the rates are not comparable. We have tried to mention these points at the end of page 2. “Strictly speaking, in Jin et al. (2020b); Rashidinejad et al. (2021)---.” However, they might not be conveyed well. We will add more detail.
>
>
> Q:The linear mixture MDP doesn't include linear MDP as a special case, which is misleading. A linear rate w.r.t. model dimension is achieved for CPPO, which is worse than the general sub-linear rate.
>
>
> A:  Thanks for pointing this out. Indeed, we do not capture the linear MDP in Jin et.al, 2020a, but it captures the linear MDP proposed by Yang & Wang, 2020. We explicitly acknowledge this fact at the beginning of Section 5:
>
>
> “Before proceeding, we clarify CPPO cannot capture linear MDPs in Jin et al. (2020a) that is different from the one (Yang & Wang, 2020) …...”
>
>
> Elaborating on the above paragraph, we answer the following. In linear MDPs [Jin et al. (2020a)], since it is not apparent how to obtain the MLE guarantee, CPPO with the log-loss might not work. Nevertheless, we can obtain the sample complexity without paying for poly |S| by modifying the log-loss to learn a model in a non-parametric way. We first can learn a $ \hat P$ as in  [Section 8.3, Agarwal et. al 2019]; then, we can get the error of $\hat P$ in terms of the distance defined as the variational form where the test functions are linear (Lemma 8.7 in Agarawal et.al 2019] . We will certainly add this detailed argument.
>
>
> Agarwal et.al (2019). Reinforcement learning: Theory and algorithms.
>
>
> Q: "computational issue": the authors should discuss the implementability. CPPO is much more impractical than prior arts.
>
>
> A: This is a good point. While we agree the current algorithm is not computationally efficient in general, this is because we tackle general MDPs requiring general function approximation while prior arts focused on tabular and linear function approximation. We add details for the rebuttal for gN7e. We respectfully disagree with the statement “CPPO is much more impractical than prior arts” since there are no prior works that can achieve the same statistical guarantee that CPPO has.
>
>
> Q:"model based vs. model free": the definitions of model based/model free are strange for me.
>
>
> We call our approach model-based because we explicitly start with a model class rather than a Q function class.  We do planning and penalization inside the learned model.
>
>
> Regarding linear MDP and tabular MDP, we agree that the difference between model-based and model-free could be blurry. We indeed cited papers like Neu & Pike-Burke (2020) which show model-free and model-based are equivalent for tabular and linear MDPs. Thus, we think we are on the same page as the reviewer for these two models. Beyond tabular and linear MDP, we do think that there is a **difference** between model-based and model-free (e.g., KNRs, factored-MDPs).
>
>
> Q: In Corollary 2, c_3 is most1/S, which seems to be considered as a constant here.
>
>
> A:  If the support of $P(\cdot |s,a)$ is compact Euclidean space, this assumption still holds (e.g. uniform distribution). Our focus is not tabular. It is just imposed to obtain the generalization bound of MLE. The same assumption is also imposed in the standard book [Corollary 14.22, Wainwright 2019].  Potentially, it can be relaxed using [van de Geer,2020].

---

> > ### Comment · Reviewer_3dYA · 2021-11-21
> > **The definition of $\pi^{\star}$ makes little difference. The comments on prior arts are unfair.**
> >
> > The definition of $\pi^{\star}$ actually makes little difference. For example, while [Rashidinejad 2021] only claims the result for optimal policy, they prove a stronger result for any deterministic expert policy (see Appendix C.5), which is the same as the standard proof for optimal policy. This reveals that "the output can only compete with the optimal policy" seems not critical for this problem. Also, you may argue that their result can only be applied for deterministic $\pi^{\star}$, but I think this difference is also not obvious. Even if you think this difference is substantial, this point is not highlighted enough for ease of comparison with prior arts, i.e., this paper is mainly devoted to generalize the assumption of deterministic $\pi^{\star}$ at the expense of performance. You should make your contribution more clear to avoid confusion.
> >
> > Given the little difference for the definition of $\pi^{\star}$, I will keep my score due to the worse performance (for tabular MDP), the stronger condition (for linear MDP), and the computational issue.

---

> > > ### Author Response · Authors · 2021-11-21
> > > **Further discussion on comparison to tabular / linear model, and the comparator policy**
> > >
> > > Thank you for your feedback and the pointer to the appendix of the work [Rashidinejad 2021].
> > >
> > > ### Main motivation and contribution of this work:
> > >
> > > First of all, we would like to emphasize our major motivation and contribution: we aim to provably **move beyond tabular and linear models**, and our main contribution is a general algorithmic framework CPPO that works for any MDPs with any function class that permits standard supervised learning MLE generalization bound. The generality of our framework is further demonstrated by instantiations on various models including KNRs, low-rank MDPs (i.e, unknown feature, thus requiring feature learning), and factored MDPs. Below is our detailed response to the reviewer’s new comment.
> > >
> > > ### Regarding S dependence in tabular MDP:
> > > We agree that our rate for tabular MDP might be suboptimal which we think is due to the fact that MLE targets the total variation distance between the learned model and the ground truth. Like prior work in the online setting such as UCRL2 [Jaksch et.al, 10], it does result in a worse rate in S dependence. However, we would like to point out that this work aims to study offline RL beyond tabular and linear models using general function approximation. We do believe generality is an important contribution, even though it comes at the cost of a worse rate in a very special model ---the tabular MDP.  Prior RL arts in both online (e.g., bellman rank and olive [Jiang et.al 17]) and offline (e.g., Xie et.al 21) that aim to move beyond tabular and linear models also come with a worse rate in S when specialized to tabular MDPs. We think improving CPPO's rate for tabular MDP **without sacrificing its generality** is an important future direction.
> > >
> > > ### Regarding computation
> > >
> > > **If we just wanted to focus on tabular MDP, then CPPO is computationally efficient** --- Similarly to UCRL2, we can implement the pessimistic planning via the Extended Value Iteration framework (sec 3.1.2 from [Jaksch et.al 10]) which was originally designed for optimistic planning in tabular MDP in UCRL2.
> > >
> > >
> > > ### Regarding linear MDP from Jin et al
> > >  We do agree at this moment the MLE does not directly work for the linear MDP model proposed by Jin et.al. However, we have demonstrated that our approach works for various other models including linear mixture model, KNRs, feature learning in low-rank MDP, and factored MDPs. Take KNRs as an example: while neither linear MDPs nor KNRs capture each other, KNRs have clear empirical evidence that they can model real world dynamical systems (e.g., see experiment results in the KRN paper, and also the assumption that a dynamical system lives in an RKHS is widely used in real world robotics papers, e.g., [Deisenroth&Rasmussen, 2011]). Thus we think it is unfair to just focus on comparison to prior algorithms that only work for linear/tabular MDPs.
> > >
> > > ### Regarding comparator policy
> > >  We do think that in offline RL setting where the offline data does not cover the global optimal policy, the ability to compete against a larger pool of policies **(including both stochastic Markovian policies and history-dependent policies)** is important. This is one of our contributions, and we did emphasize it clearly in section 4 right after theorem 1 -- this is one of the important points that distinguish ours from the prior art [Xie et.al 21]. We will emphasize this point more in a revised version.
> > >
> > > ### References:
> > > [Jaksch et.al, 10] Near-optimal Regret Bounds for Reinforcement Learning, JMLR 2010
> > >
> > > [Deisenroth&Rasmussen, 2011], PILCO: A Model-Based and Data-Efficient Approach to Policy Search, 2011 ICML

---

> > > > ### Author Response · Authors · 2021-11-22
> > > > **Further discussion on the tabular case (Difference of our result and [Rashidinejad et.al 21])**
> > > >
> > > > We thank the reviewer again for pointing us to appendix C.5 of the work [Rashidinejad et.al21]. We double-checked the claim there, and to the best of our knowledge, the result there implies
> > > >
> > > > "The learned policy can compete with all deterministic policies in expectation with $\sqrt{|S|}$ dependence”.
> > > >
> > > > Our result implies
> > > >
> > > > “The learned policy can compete with **all** stochastic policies (including history-dependent ones) in high probability with $|S|$ dependence”.
> > > >
> > > > Hence, the difference is not only **deterministic** versus **stochastic**, but also **in expectation** versus **in high probability**. It is unclear to us whether the work from Rashidinejad et.al 21 can obtain such a high probability statement that holds simultaneously for all policies. In fact, other papers like [Zannete et.al21] and [Xie et.al 21], which obtain our type guarantee, imply $|S|$ dependence. We will definitely add this crucial comparison to [Rashidinejad et.al 21] !

---

### Author Response · Authors · 2021-11-12
**Summary of rebuttal**

We would like to thank the reviewers for their comments. In summary, reviewers mainly bring up the concern regarding (1) superiority over prior works, (2) computational efficiency.

We have answers as follows.

* Regarding (1), we emphasize that we mainly focus on answering the challenging question on **whether we can information-theoretically learn under partial coverage with general function approximation**. Prior works mainly consider **special models** with tabular or linear structures.

* Regarding (2), because of the generality (i.e., considering any models beyond tabular and linear models), the computational efficiency of the proposed algorithm is indeed sacrificed. Still, we also have mentioned a possible direction to gain more computational efficiency.

---

### Decision · Program_Chairs · 2022-01-20

**Decision:**

Accept (Poster)

**Comment:**

In this paper, the authors consider the offline RL with only realizability and partial coverage assumption, under which a model-based pessimistic policy optimization algorithm has been proposed and rigorously justified. Moreover, variety of special MDP models, including kernelized nonlinear regulator and linear mixture MDP, have been plugged into the general framework, which leads to different specific algorithm and refined guarantees.

In general, the reviewers are positive to the submission. However, there are still issues need to be further discussed,

- *Computation feasibility*: most of the reviewers raise the same concern about the computation feasibility and efficiency. Specifically, the proposed algorithm is too complicated, and thus, may not be practical.

- *Comparison with existing statistical results*: both reviewers and I appreciate the summary in the paper about the coverage assumptions in the existing methods. However, a similar table for summarizing the complexity of existing algorithms, as well as detailed discussion, is also necessary for a better position of the proposed method among the literature, including both model-based and model-free RL.